# FAT10 is phosphorylated by IKKβ to inhibit the antiviral type-I interferon response

Kritika Saxena[1,*], Nicola Domenico Roverato[1,*], Melody Reithmann[1], Mei Min Mah[1], Richard Schregle[1,2], Gunter Schmidtke[1], Ivan Silbern[3,4] ⓘ, Henning Urlaub[3,4], Annette Aichem[1,2] ⓘ

**IFN-I secretion provides a rapid host defense against infection with RNA viruses. Within the host cell, viral RNA triggers the activation of the RIG-I signaling pathway, leading to the production of IFN-I. Because an exaggerated IFN-I response causes severe tissue damage, RIG-I signaling is tightly regulated. One of the factors that control the IFN-I response is the ubiquitin-like modifier FAT10, which is induced by TNF and IFNγ and targets covalently FAT10-linked proteins for proteasomal degradation. However, the mechanism of how FAT10 modulates IFN-I secretion remains to be fully elucidated. Here, we provide strong evidence that FAT10 is phosphorylated by IκB kinase β (IKKβ) upon TNF stimulation and during influenza A virus infection on several serine and threonine residues. FAT10 phosphorylation increases the binding of FAT10 to the TRAF3-deubiquitylase OTUB1 and its FAT10-mediated activation. Consequently, FAT10 phosphorylation results in a low ubiquitylation state of TRAF3, which is unable to maintain interferon regulatory factor 3 phosphorylation and downstream induction of IFN-I. Taken together, we reveal a mechanism of how phosphorylation of FAT10 limits the production of tissue-destructive IFN-I in inflammation.**

## Introduction

The innate immune system represents the host´s first-line defense against viral infections (Koyama et al, 2008). In this context, IFN-I are produced by all nucleated cells and represent the principal cytokines that counteract viral replication (Ivashkiv & Donlin, 2014). In fact, several receptors of the infected cell can recognize the genetic material of the virus to elicit an immune response. For instance, the endosomal TLR3 recognizes double-stranded (ds) viral RNA, which initiates a signal transduction cascade leading to IFNα/β production (Vercammen et al, 2008). The viral RNA is also detected by cytoplasmic sensors named RIG-I-like receptors (RLRs). Among them, RIG-I (retinoic acid inducible gene 1) binds tri-phosphorylated single-stranded (ss) RNA and short dsRNA, whereas melanoma differentiation-associated protein 5 (MDA5) preferentially recognizes longer dsRNA (Kato et al, 2006). Once activated by their ligand, these two cytosolic receptors associate with MAVS (mitochondrial antiviral-signaling protein), leading to IFN-I secretion (Kawai et al, 2005). Notably, both the activation of RIG-I/melanoma differentiation-associated protein 5 and of TLR3 receptors enhance the enzymatic functions of the E3 ligases TRAF3 and TRAF6 (Häcker et al, 2006; Vercammen et al, 2008; Mao et al, 2010; Liu et al, 2013a). TRAF3 auto-ubiquitylates itself with K63-linked ubiquitin chains to activate the kinase complex TBK1/IKKε ([TANK]-binding kinase 1/inducible IκB kinase ε) (Fitzgerald et al, 2003; Häcker et al, 2011), which in turn phosphorylates and activates the transcription factor interferon regulatory factor 3 (IRF3) in the cytoplasm. Phosphorylated IRF3 then translocates into the nucleus where it binds to the promotor of IFN-I genes leading to their transcriptional activation (Fitzgerald et al, 2003; Sharma et al, 2003). TRAF6 is also auto-modifying itself with K63-linked ubiquitin chains, but this event instead leads to the activation of the IKK kinase complex (Ikkα/IKKβ/ IKKγ) (Lamothe et al, 2007), ultimately triggering NF-κB and IRF7 activation (Konno et al, 2009). The ubiquitin–proteasome system was described to fine-tune the IFN-I signaling pathway. For instance, the two ubiquitin-ligases TRIM25 and RIPLET poly-ubiquitylate RIG-I with K63-linked ubiquitin chains during viral infection to enhance the formation of the RIG-I–MAVS complex (Gack et al, 2007; Gack et al, 2008; Oshiumi et al, 2009). On the other hand, the same pathway can be negatively regulated to avoid tissue damage and chronic inflammatory diseases. The deubiquitylating enzyme OTUB1 (otubain-1), for example, translocates to the mitochondrion during viral infection to promote the K63 deubiquitylation of TRAF3 and TRAF6 (Li et al, 2010), whereas it simultaneously acts as an inhibitor of the proteasomal elimination of RIG-I (Jahan et al, 2020). Ubiquitin-like modifiers (ULMs) have also been reported to influence the IFN-I signaling pathway (Liu et al, 2013b). HLA-F adjacent transcript 10 (FAT10) is an 18-kD protein of the ULM family that is basally expressed in organs of the immune system but can also be induced in virtually

---

[1]Department of Biology, Division of Immunology, University of Konstanz, Konstanz, Germany  [2]Biotechnology Institute Thurgau at The University of Konstanz, Kreuzlingen, Switzerland  [3]Bioanalytical Mass Spectrometry Research Group, Max Planck Institute for Multidisciplinary Sciences, Göttingen, Germany  [4]Bioanalytics, Institute for Clinical Chemistry, University Medical Center Göttingen, Göttingen, Germany

Correspondence: Annette.Aichem@bitg.ch
*Kritika Saxena and Nicola Domenico Roverato contributed equally to this work

every tissue by a synergistic stimulation with the pro-inflammatory cytokines TNF and IFNγ (Lukasiak et al, 2008; Schregle et al, 2018). FAT10 consists of two ubiquitin-like (UBL) domains that are joined by a flexible linker (Aichem et al, 2018; Aichem & Groettrup, 2020), and it is the only ULM which directly targets substrate proteins for proteasomal degradation independently of ubiquitin (Hipp et al, 2005; Schmidtke et al, 2009). FAT10 has recently been reported to downregulate the IFN-I response (Nguyen et al, 2016; Zhang et al, 2016; Mah et al, 2019; Wang et al, 2019). In fact, FAT10-deficient mice elicit an enhanced IFN-I secretion during infection with lymphocytic choriomeningitis virus (Mah et al, 2019), suggesting that FAT10 can negatively regulate the antiviral response. From a mechanistic perspective, FAT10 expression was reported to be induced during influenza A virus (IAV) infection (Zhang et al, 2016). In addition, FAT10 was shown to either sequester RIG-I into an insoluble compartment (Nguyen et al, 2016), or to impair the activating K63-linked ubiquitylation of RIG-I through its non-covalent interaction with the ubiquitin E3 ligase ZNF598 (Wang et al, 2019). Moreover, FAT10 was shown to stimulate the stability and the enzymatic activity of OTUB1 in reducing the K63- and K48-linked poly-ubiquitylation of TRAF3 (Bialas et al, 2019), suggesting that FAT10 can alter this pathway at different stages. In this work, we show that FAT10 is phosphorylated at multiple sites by IKKβ, both, upon TNF stimulation and during IAV infection. We also provide strong evidence that phospho-FAT10 (p-FAT10) formation is essential for the efficient inhibition of IFN-β secretion, and that this effect relies on the capacity of p-FAT10 to efficiently bind to OTUB1 to decrease the overall ubiquitylation and IFN-I induction via TRAF3.

## Results

### FAT10 is phosphorylated at multiple sites

The discovery that ubiquitin is phosphorylated by PINK1 during mitophagy provided valuable insights into how phosphorylation of a ubiquitin family modifier can modulate the function of the ubiquitin–proteasome system with significant impact on cellular homeostasis (Kane et al, 2014; Kazlauskaite et al, 2014; Koyano et al, 2014; Lazarou et al, 2015). The phosphorylation of ubiquitin at Ser65 causes major structural changes, with consequences for its biochemical and functional properties (Wauer et al, 2015a, 2015b; Dong et al, 2017). Inspired by these reports, we aimed to study whether FAT10 is modified by phosphorylation as well, and whether this could modulate its activity in cells. Accordingly, we induced FAT10 expression by treating HEK293 cells with TNF and IFNγ. After 24 h, cells were lysed in the presence of a phosphatase inhibitor, and the FAT10 protein was immunoprecipitated with a highly specific monoclonal FAT10-reactive antibody (Fig 1A). After SDS–PAGE, the gel fractions containing FAT10 were cut out and subjected to a phospho-proteomic analysis. Successful immunoprecipitation of FAT10 was further confirmed by immunoprecipitation (IP)/immunoblotting (IB) analysis (Fig 1A, right panel). In parallel, we performed a similar experiment with overexpressed 6His-3xFLAG-FAT10 (named hereafter as FLAG-FAT10), however, in this case, in the absence of TNF. Remarkably, we found that FAT10 is phosphorylated at five different sites: Ser62, Ser64, Thr77, Ser95, Ser109 (schematically shown in Fig 1B), with both endogenous and FLAG-tagged FAT10s displaying a similar phosphorylation profile (Table S1, see Data Availability information for access to original MS data files). Furthermore, after we showed that the MAPKAPK3 kinase (MK3) is able to phosphorylate FAT10 in vitro by a radiometric-filter plate assay (Fig S1), we measured the incorporation of radioactive ATP into recombinant FAT10 by MK3 and by PINK1. Using autoradiography, we found that FAT10 is phosphorylated by MK3 but not by PINK1 (Fig 1C, lanes 8–11), suggesting that the two kinases have different affinities and specificities for ubiquitin and FAT10.

### FAT10 phosphorylation is mediated by TNF

As our phospho-mass spectrometric approach indicated that FAT10 is phosphorylated both, under endogenous conditions in TNF/IFNγ-stimulated HEK293 cells, and in the absence of cytokines, we aimed to corroborate this notion by using a combined immunoprecipitation (IP)/immunoblot (IB) approach. Hence, HEK293 cells were transiently transfected with a FLAG-FAT10 encoding plasmid for 24 h. Subsequently, cells were treated with TNF/IFNγ for 24 h or transfected with different plasmids encoding distinct forms of the FAT10 in vitro-kinase MK3 (HA-MK3 TT/EE and HA-MK3 TT/AA, representing the constitutive active and inactive MK3 forms, respectively [Ludwig et al, 1996]). Where indicated, cells were additionally treated with starvation/TPA (PMA) stimulation to induce MK3 enzymatic activity (Ludwig et al, 1996). 24 h later, cells were lysed, and where indicated, the lysate was exposed to CIP (calf intestinal alkaline phosphatase) treatment. Subsequently, an anti-phosphoserine antibody was used to pull-down serine-phosphorylated proteins, followed by SDS–PAGE/IB analysis. Interestingly, we found that phosphorylation of FLAG-FAT10 was enhanced by TNF/IFNγ, and that this modification was strongly reversed by CIP treatment (Fig 2A, lanes 4 and 5). However, neither MK3 overexpression, nor starvation/TPA treatment could enhance the phosphorylation of FAT10 under in cellulo conditions (Fig 2A, lanes 6–9). To confirm this result, we transfected HEK293 cells with a FLAG-FAT10 expression plasmid and performed a Phos-tag gel coupled to SDS–PAGE/IB analysis, because this is a well-known method to study protein phosphorylation (Kinoshita et al, 2009; Shiba-Fukushima et al, 2012; Kane et al, 2014). In addition, to show phosphorylation of FAT10 also under endogenous conditions, we induced endogenous FAT10 expression by treating HEK293 cells for 24 h with TNF/IFNγ. As shown in Fig 2B, it was confirmed that TNF/IFNγ treatment enhanced phosphorylation of FLAG-FAT10 and likewise also that of endogenous FAT10 (Fig 2B, lanes 4 and 7). Because our phospho-proteomic analysis had revealed that FLAG-FAT10 was phosphorylated already in the absence of TNF/IFNγ, we suggest that a basal phosphorylation of FAT10 must exist which is further enhanced by TNF/IFNγ treatment. Of note, the portion of FAT10 which becomes phosphorylated was estimated to be ~5% of the total FAT10 protein amount. To see which of both stimuli, TNF or IFNγ, promotes FAT10 phosphorylation, we stimulated FLAG-FAT10 expressing cells with either TNF or IFNγ. We found that TNF, but not IFNγ, is the cytokine that triggers the phosphorylation of FAT10 (Fig 2C, lanes 3 and 4). Notably, the TNF stimulation by itself slightly

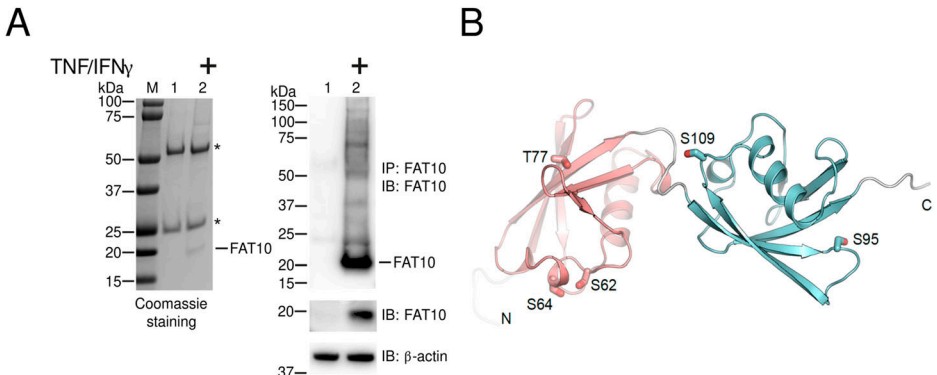

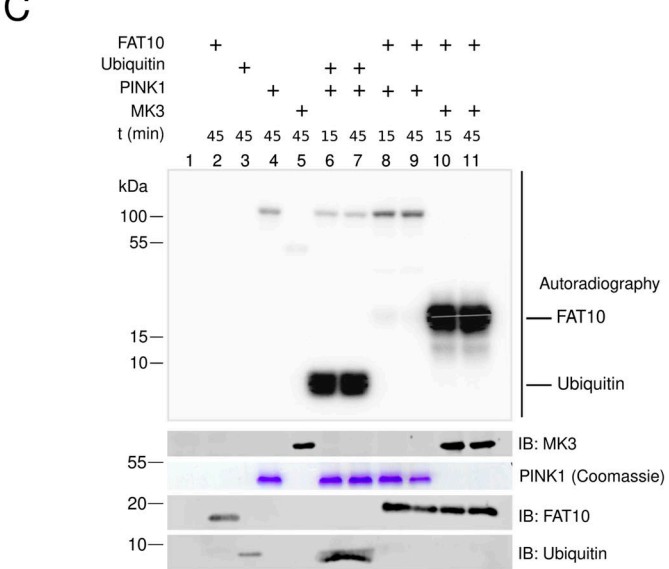

**Figure 1. FAT10 is phosphorylated at multiple sites.**

**(A)** The expression of endogenous FAT10 was stimulated by treating HEK293 cells with TNF/IFNγ for 24 h, followed by immunoprecipitation (IP) with a monoclonal anti-FAT10 antibody (4F1), SDS–PAGE and Coomassie blue staining (left panel). Endogenous FAT10 was cut out and sent for a phospho-proteomic analysis. As a control, samples were additionally analyzed by immunoblotting (IB) (right panel). Endogenous FAT10 was visualized with a FAT10-reactive, rabbit polyclonal antibody (Hipp et al, 2005). β-Actin was used as a loading control. Asterisks mark the heavy and light chains of the FAT10-reactive antibody used for the immunoprecipitation. **(B)** Ribbon diagram of FAT10 showing the phosphorylated amino acids Ser62, Ser64, Thr77, Ser95, and Ser109 in the N- (red) and C- (blue) ubiquitin-like domain, respectively. **(C)** Radiolabeled phosphate was incorporated into recombinant FAT10 or ubiquitin during incubation with the recombinant kinases PINK1 or MAPKAPK3 (MK3) at 30°C for 15 or 45 min. The autoradiogram shows substrate specificity of MAPKAPK3 and PINK1 to phosphorylate FAT10 and ubiquitin, respectively. One representative experiment out of three independent experiments with similar outcomes is shown.
Source data are available for this figure.

increased FLAG-FAT10 expression (Fig 2C, lanes 2 and 3 versus 4 and 5) which is because of the fact that TNF activates the CMV promoter which is upstream of the FLAG-FAT10 gene in the expression plasmid (Stein et al, 1993). Based on the detection of several bands we observed in the upper part of the blot, we supposed that the FAT10 monomer might be phosphorylated in a differential manner, or that phosphorylated FAT10 might be conjugated to specific substrates after phosphorylation. Overall, the data in Fig 2 suggest that a fraction of FAT10 is phosphorylated and that TNF induces this posttranslational modification (PTM).

### IKKβ phosphorylates FAT10

To identify the kinase(s) responsible for the phosphorylation of FAT10, we used a "by-exclusion" approach using specific inhibitors against the family of kinases that are known to be directly or indirectly activated by TNF. Hence, we overexpressed FLAG-FAT10 in HEK293 cells and treated them for 24 h with TNF. Where indicated in Fig 3A, cells were pretreated with the indicated kinase inhibitors (3 h before TNF treatment). FAT10 phosphorylation was evaluated by FLAG-IP/Phos-tag/SDS–PAGE/IB analysis, as described afore. We

found that FAT10 phosphorylation was strongly reduced by a pan-IKK kinase inhibitor (IKK-16), and partially by a pan-JNK kinase inhibitor (SP600125) (Fig 3A, lanes 4 and 6). Consequently, we studied the capacity of the single members of the two families of kinases to phosphorylate recombinant FAT10. We performed an IP/kinase assay where we overexpressed the indicated FLAG- or HA-tagged kinases in HEK293 cells, including also the ones that are not expressed in HEK293 cells (i.e., IKKε) (Fig 3B). After immunoprecipitation of the kinases, beads were washed and an in vitro kinase assay with recombinant FAT10 was performed directly on the beads, followed by Phos-tag/SDS–PAGE/IB analysis. Remarkably, we found that the two members of the IKK family, namely IKKβ and IKKε, strongly induced FAT10 phosphorylation, and that JNK3 only partially induced this PTM (Fig 3B, lanes 5, 7 and 8). To confirm these results, we overexpressed the three kinases in HEK293 cells along with FLAG-FAT10. Subsequently, we treated the cells for 24 h with TNF, lysed them, and performed a FLAG-IP/Phos-tag/SDS–PAGE/IB analysis (Fig 3C). We observed that the co-expression of FLAG-FAT10 together with IKKβ, IKKε, and JNK3 strongly induced FAT10 phosphorylation (Fig 3C, lanes 4, 5, and 7). Interestingly, the three kinases gave rise to different phosphorylation profiles of FAT10, and their

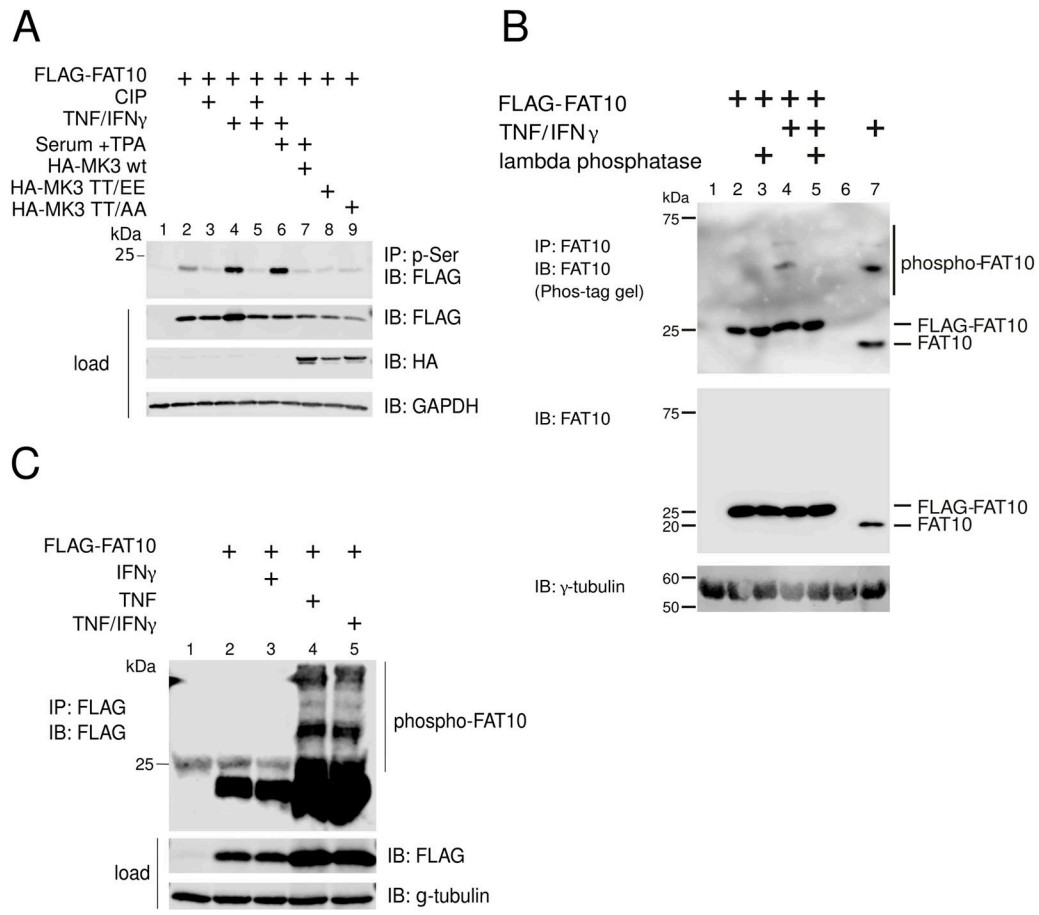

**Figure 2. Phosphorylation of FAT10 upon TNF stimulation.**
**(A)** His-3xFLAG-FAT10 (FLAG-FAT10), HA-tagged MK3, HA-MK3 TT/EE (constitutively active mutant), or HA-MK3 TT/AA (inactive mutant), were transiently overexpressed in HEK293 cells for 24 h, followed by lysis and immunoprecipitation (IP) with a monoclonal phosphoserine-reactive antibody. Subsequently, an immunoblot (IB) was performed using the antibodies indicated. Where indicated, cells were stimulated with TNF/IFNγ for 24 h before harvesting and lysis. Moreover, cells were starved for 24 h (0.3% FCS DMEM) followed by TPA treatment (30 min) before lysis, where indicated. Calf intestinal alkaline phosphatase phosphatase was added to the lysates used in lanes 3 and 5, 2 h before performing the immunoprecipitation. **(B)** HEK293 cells were transiently transfected with an expression plasmid for FLAG-tagged FAT10 and where indicated, additionally treated with TNF/IFNγ for 24 h. Endogenous FAT10 expression was induced by treating HEK293 cells with TNF/IFNγ for 24 h. Where indicated, lysates were incubated with 400 U of λ phosphatase for 30 min at 30°C, before the immunoprecipitation was performed. Subsequently, an immunoprecipitation against FAT10 was performed using a monoclonal FAT10-reactive antibody (clone 4F1, [Aichem et al, 2010]) coupled to protein A sepharose, followed by Phos-tag/SDS–PAGE and IB analysis with the antibodies indicated. γ-tubulin was used as loading control. **(C)** Cells were prepared as in (A) and treated as specified, followed by FLAG-IP, Phos-tag/SDS–PAGE, and IB analysis with the indicated antibodies. One representative example out of three independent experiments with similar outcomes is shown. Source data are available for this figure.

co-expression showed a synergistic effect in phosphorylating FAT10 (Fig 3C, lanes 6, 9 and 10). Based on the in vitro data shown in Fig 3B, showing that JNK3-mediated FAT10 phosphorylation was only modest, and based on the fact that the JNK3 expression profile reportedly is limited to the nervous system (Yoshitane et al, 2012), we pursued the investigation of the IKKβ- and IKKε-mediated phosphorylation of FAT10. Thus, we asked whether these two kinases could modify FAT10 under in vitro conditions (Fig 3D). We performed an in vitro kinase assay using recombinant FAT10 (rFAT10), rIKKβ, rIKKε, and rMK3, (the latter was used as a positive control), and found that both, IKKβ and IKKε, led to the formation of a highly phosphorylated form of FAT10, though not quantitatively (as seen in case of MK3) (Fig 3D, lanes 3, 4, and 5). Finally, to understand whether both kinases, IKKβ and IKKε, are required for FAT10 phosphorylation in cells, we used A549 cells, which is a

human alveolar adenocarcinoma cell line that expresses both endogenous IKKβ and IKKε (Matikainen et al, 2006; Huang et al, 2007). After FLAG-FAT10 overexpression, cells were stimulated with TNF and/or treated with a specific IKKβ inhibitor (TPCA-1) or with a specific IKKε inhibitor (CAY10576). By FLAG-IP/Phos-tag/SDS–PAGE/IB analysis, we found that the IKKβ inhibitor, but not the IKKε inhibitor, caused a prominent reduction of FAT10 phosphorylation (Fig 3E, lane 4). This result suggests that IKKβ acts as the major cellular kinase for FAT10 phosphorylation.

## IKKβ stimulates FAT10 phosphorylation, but not FAT10 expression, upon IAV infection

As FAT10 was reported to be expressed during IAV infection through the IKKβ-dependent activation of NF-κB (Zhang et al, 2016), we

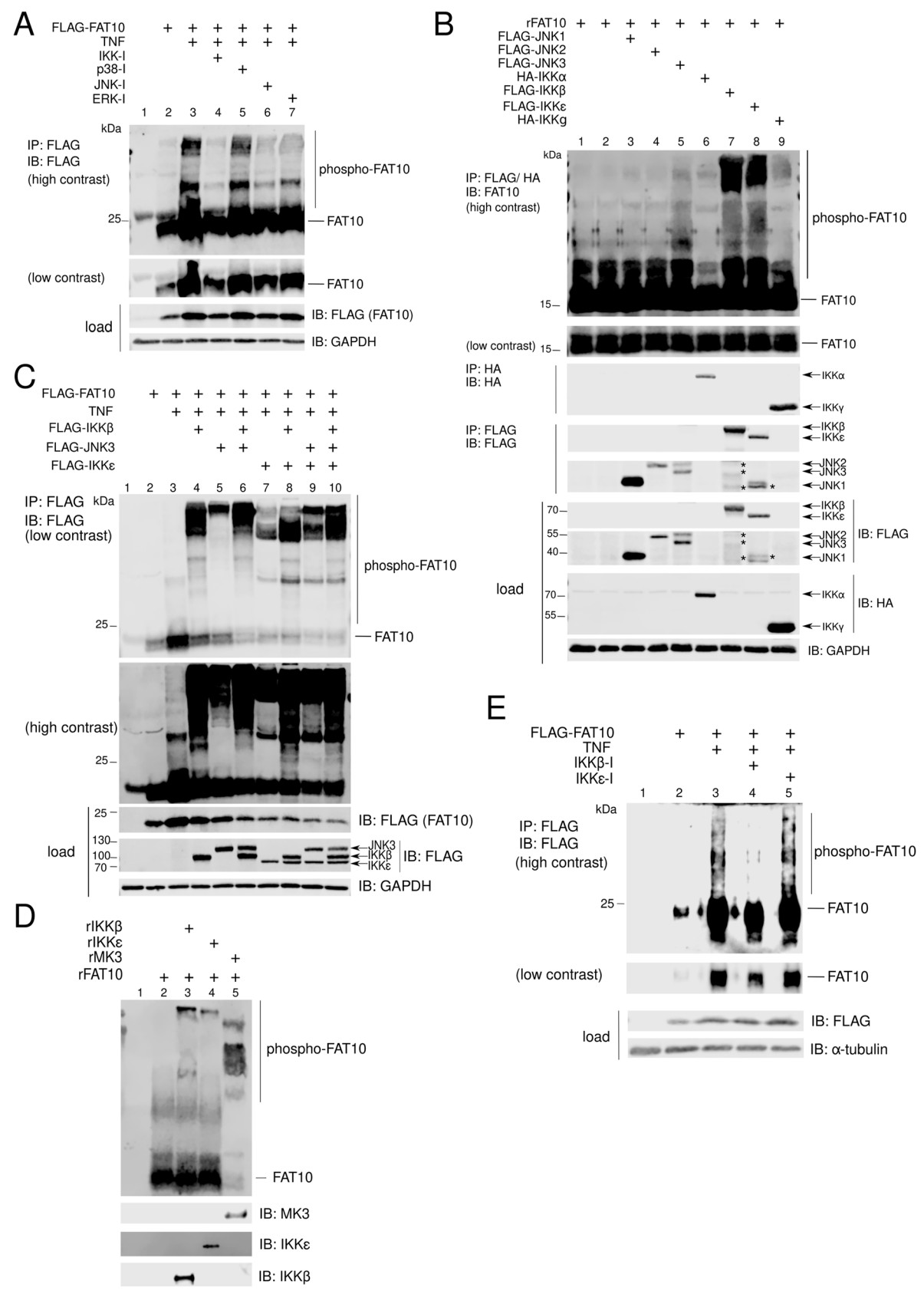

aimed to investigate whether the virus-mediated activation of IKKβ could lead to the expression and/or to the phosphorylation of FAT10. Therefore, we stimulated A549 cells either with TNF, or infected them with IAV. 1 d later, we collected the cells and performed a real-time PCR analysis for FAT10 expression. As expected, FAT10 expression was strongly induced by TNF/IFNγ (Fig 4A), as previously reported (Raasi et al, 1999; Lukasiak et al, 2008). However, IAV infection did not induce any significant transcription of the FAT10 gene (Fig 4A). To use an alternative readout, we performed an immunoblot analysis of IAV-infected A549 cells at different time points. We noticed a significant induction of RIG-I expression upon IAV infection, which was accompanied by the degradation of OTUB1 and by an elevation in the levels of the IAV viral matrix 1 (M1) protein (Fig 4B), as previously reported (Jahan et al, 2020). However, also under these experimental conditions, no FAT10 protein expression was detectable upon IAV infection. To further confirm this finding, we performed an immunoprecipitation of the entire pool of endogenous FAT10 from TNF/IFNγ-stimulated, or IAV-infected A549 cells by using a high affinity monoclonal FAT10-reactive antibody (clone 4F1, [Aichem et al, 2010]). After immunoblot analysis, we could confirm that the synergistic treatment with the proinflammatory cytokines TNF and IFNγ is the only stimulus inducing FAT10 expression, and that IAV infection alone is not sufficient to induce FAT10 expression in A549 cells (Fig 4C). Nonetheless, we suggested that the activation of IKKβ that takes place during an IAV infection (Meylan et al, 2005) would promote FAT10 phosphorylation. To investigate this hypothesis, we overexpressed FAT10 in A549 cells and stimulated them with TNF, or infected them with IAV. 1 d later, cells were lysed and a FLAG-IP/Phos-tag/SDS–PAGE/IB analysis was performed (Fig 4D). Remarkably, we found that not only TNF induced the phosphorylation of FAT10, but that also the IAV infection could boost this PTM (Fig 4D). Next, we aimed to assess whether this effect was because of TNF that was secreted during an IAV infection, or if this was due to the activation of the RIG-I/TLR3 signaling pathway. Hence, we first measured the level of TNF in the supernatant of IAV-infected cells and found that there was no significant increase in TNF secretion 24 h after IAV infection (Fig S2). Second, we found that the cell lysate from IAV-infected A549 cells that was incubated with recombinant FAT10 could induce FAT10 phosphorylation (Fig 4E, lane 2), suggesting that FAT10 can be phosphorylated during IAV infection, independent of TNF. Lastly, because IAV infection is known to trigger the activation of both, IKKβ and IKKε (Pham & TenOever, 2010), we investigated which kinase was responsible for FAT10 phosphorylation upon IAV infection.

Accordingly, we infected A549 cells with IAV and performed a FLAG-IP/Phos-tag/SDS–PAGE/IB analysis (Fig 4F). Here, we observed that the selective inhibition of IKKβ, but not the inhibition of IKKε, decreased FAT10 phosphorylation, suggesting that IKKβ acts as a FAT10 kinase upon its IAV-mediated activation (Fig 4F, lanes 4 and 5).

## FAT10 phosphorylation does not alter the key biochemical properties of FAT10

With the aim to further characterize the biological outcomes of FAT10 phosphorylation, we performed a phospho-mimetic analysis. We mutated the five phosphorylation sites of FLAG-tagged FAT10 either to negatively charged amino acid residues (S62E, S64E, T77D, S95E, S109E) to mimic the negative charge of the phosphate group (phospho-mimicking, named hereafter as FLAG-FAT10 E) or to alanine (S62A, S64A, T77A, S95A, S109A) to impede FAT10 phosphorylation (phospho-deficient, named hereafter as FLAG-FAT10 A). As proof-of-principle, we investigated the phosphorylation status of the FAT10 phospho-mutants upon TNF stimulation. We stably expressed FLAG-FAT10 WT, FLAG-FAT10 E, and FLAG-FAT10 A in A549 cells and performed an immunoprecipitation against the FLAG-tag. Phosphorylation of the FAT10 variants was analyzed by a Phos-tag/SDS–PAGE/IB analysis with FLAG-reactive antibodies. Notably, the mutation of these sites to alanine clearly reduced phosphorylation of FAT10 A (Fig 5A, lanes 2–5 and Fig 5B). In case of FAT10 E, the phosphorylation status could not be determined because a mutation to glutamic acid (E) already causes a retardation of proteins in Phos-tag gels (Fig 5A and quantification of ECL signals in Fig 5B). FAT10 has a high tendency to precipitate (Aichem et al, 2018). Therefore, we tested if the introduction of the mutations might influence the solubility of FAT10 and analyzed the appearance of FAT10 in soluble and insoluble fractions of the cells. However, we found that these mutations did not lead to destabilization of the FAT10 structure because we did not observe a significant increase of the FAT10 mutants in the insoluble fraction, as compared with WT FAT10 (Fig S3). Next, we investigated whether the phospho-mutants might possess altered biochemical characteristics. We first overexpressed the different forms of FAT10 in HEK293 cells, followed by FLAG-IP/IB analysis and saw that the introduced mutations did not affect FAT10 bulk conjugation to substrate proteins (Fig 5C). Moreover, the two phospho-mutants showed the same kinetic of proteasomal degradation during a cycloheximide (CHX) chase assay (Fig 5D and quantification of ECL signals in Fig 5E). Lastly, because protein phosphorylation can alter the cellular localization

**Figure 3. FAT10 is phosphorylated by IKKβ.**
**(A)** HEK293 cells were transiently transfected with a His-3xFLAG-FAT10 (FLAG-FAT10) expression construct and stimulated for 24 h with TNF. Lysates were subjected to immunoprecipitation using FLAG-reactive antibodies, coupled to sepharose beads, and subsequently analyzed by Phos-tag/SDS–PAGE/IB analysis. Where indicated, cells were pretreated before TNF stimulation with the displayed inhibitors for a total of 3 h (10 μM each). **(B)** HEK293 cells were transiently transfected with expression plasmids for the different kinases. Cells were harvested, lysed, and subjected to immunoprecipitation using anti-FLAG or anti-HA antibodies, coupled to sepharose beads. Subsequently, the immunoprecipitated kinases were incubated with recombinant FAT10 (rFAT10) and an in vitro reaction was performed in the kinase buffer. The phosphorylation status of FAT10 was analyzed by Phos-tag/SDS–PAGE and IB. Asterisks mark unspecific background bands. **(C)** FLAG-FAT10 and the indicated kinases were transiently overexpressed in HEK293 cells followed by TNF stimulation. After 24 h, cells were lysed and subjected to immunoprecipitation against the FLAG-tag, combined with Phos-tag/SDS–PAGE and IB analysis. **(D)** Recombinant FAT10 (rFAT10) was incubated with recombinant kinases IKKβ, IKKε or MK3 for 45 min at 30°C. Subsequently, proteins were separated on a Phos-tag/SDS–PAGE followed by immunoblot analysis using the antibodies indicated. **(E)** HEK293 cells were prepared as described in (A). Where specified, cells were pretreated with the inhibitors indicated (10 μM each) for a total of 3 h before stimulation with TNF. One representative example out of three independent experiments with the same outcomes is shown.
Source data are available for this figure.

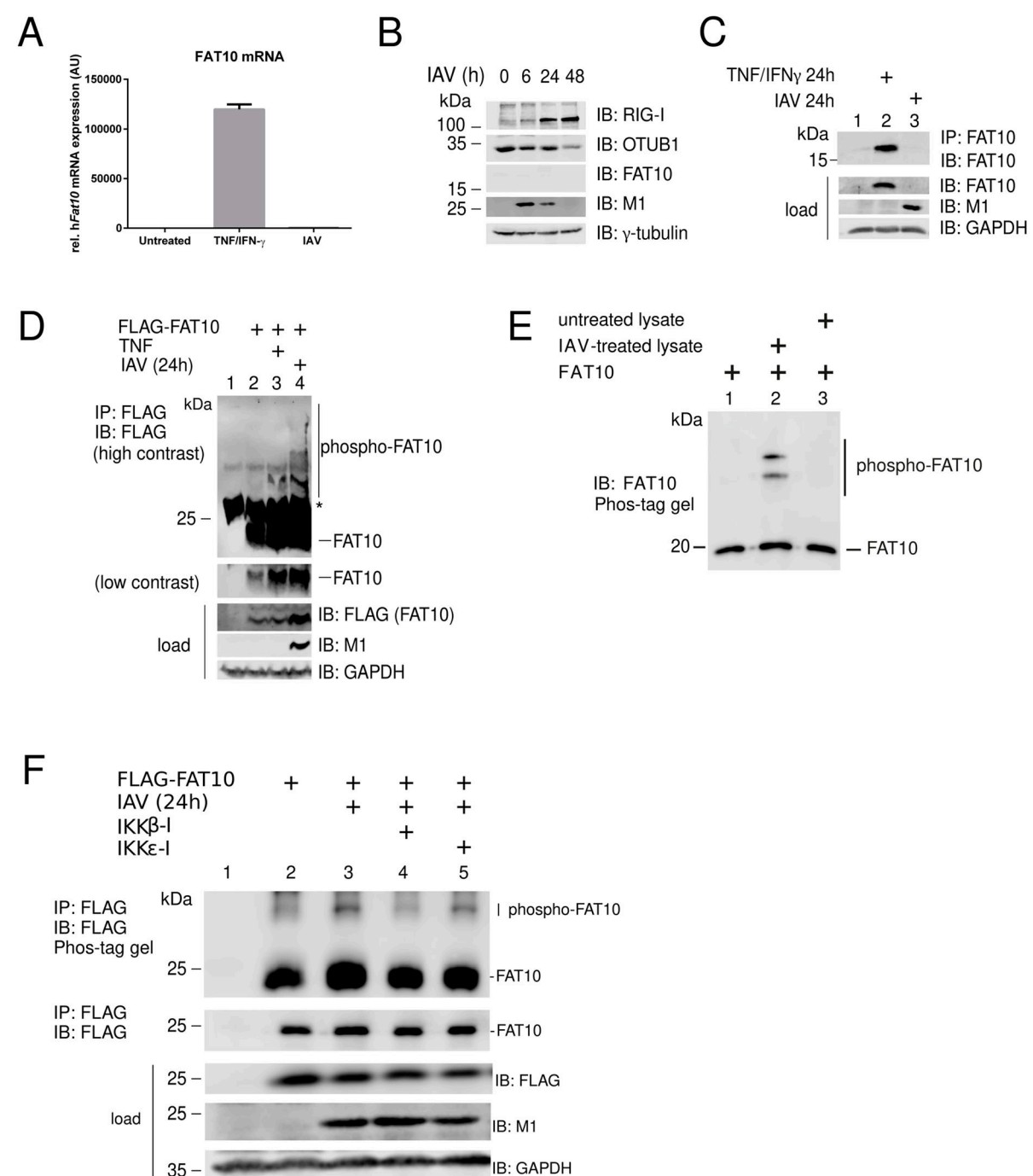

**Figure 4. Influenza A virus (IAV) infection induces FAT10 phosphorylation.**
**(A)** FAT10 mRNA levels were analyzed and quantified by real-time PCR in A549 cells 24 h after TNF/IFNγ treatment or 24 h after IAV infection. **(B)** A549 cells were infected with IAV (MOI:1) for 0, 6, 24 or 48 h, as indicated. Cell lysates were subjected to SDS–PAGE combined with immunoblot analysis with the antibodies indicated. **(C)** Immunoblot showing endogenous FAT10 expression in A549 cells treated for 24 h with TNF/IFNγ, or infected for 24 h with IAV. FAT10 was immunoprecipitated with a monoclonal FAT10-reactive antibody (clone 4F1) and an immunoblot was performed with the antibodies indicated. **(D)** A549 cells stably expressing His-3xFLAG-FAT10 (FLAG-FAT10) were lysed after 24 h of TNF stimulation or after 24 h of IAV infection. The lysates were subjected to immunoprecipitation using FLAG-reactive antibodies coupled to sepharose beads. Subsequently, a Phos-tag/SDS–PAGE and immunoblot analysis with the indicated antibodies was performed. Asterisk marks an unspecific background band. **(E)** Recombinant 6His-SUMO-FAT10 was purified via Ni-NTA and left bound to the beads. 400 units λ phosphatase was added and incubated twice for 30 min at 30°C. Beads were extensively washed and FAT10 was eluted by treatment with the ULP1 enzyme to receive untagged and non-phosphorylated FAT10. Lysates from untreated or IAV-infected A549 cells were prepared and incubated with the purified FAT10, as indicated, for 30 min at 30°C. Proteins were separated on a Phos-tag/SDS–PAGE and analyzed by immunoblotting with a FAT10-reactive, polyclonal antibody. **(F)** A549 cells stably expressing FLAG-FAT10 were infected with IAV for 24 h. Cell lysates were subjected to a FLAG-immunoprecipitation and analyzed by Phos-tag/SDS–PAGE, followed by immunoblotting using the antibodies indicated. Where indicated, cells were pretreated with the indicated inhibitors (5 mM IKKβ inhibitor, 10 μM IKKε inhibitor) for 3 h, before IAV infection. For each panel, one representative example out of three independent experiments with similar outcomes is shown.
Source data are available for this figure.

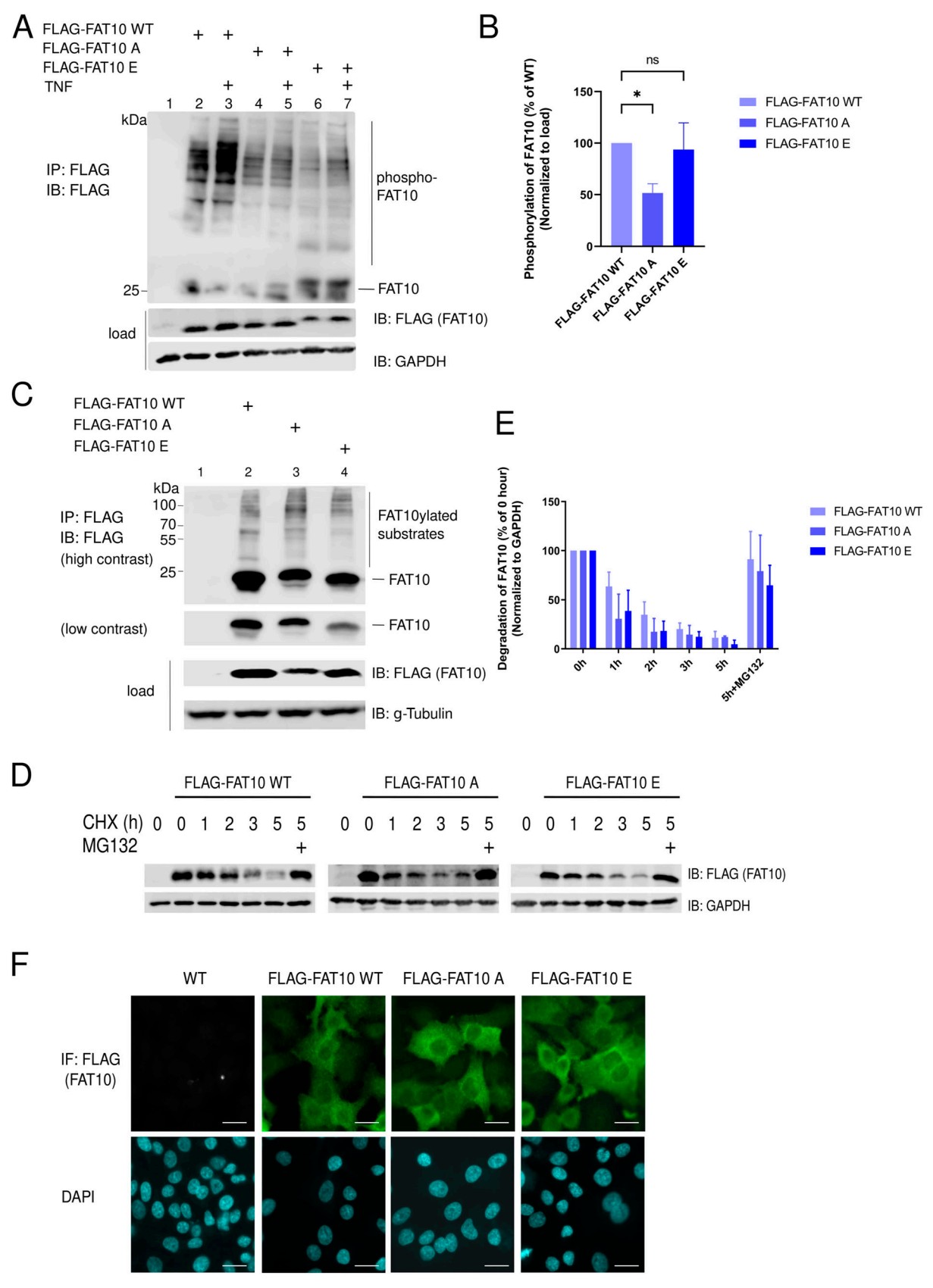

of proteins (Day et al, 2016), we aimed to study the subcellular localization of FAT10 and the FAT10 phospho-mutants. Also in this case, we found no significant alteration of the subcellular localization of the FAT10 variants, which were distributed both, in the cytosolic and the nuclear compartments, independent of the mutations of the phosphorylation sites (Fig 5F). Thus, we concluded, that the introduced mutations to generate a phosphorylation mimicking or dead FAT10 mutant did not alter principle functions such as conjugation capability or subcellular localization of the respective FAT10 mutant.

**Phosphorylation is required for the FAT10-mediated inhibition of IFN-I secretion**

FAT10 has been described to inhibit the secretion of IFN-$\beta$ in vivo and in several cellular models (Mah et al, 2019). Considering that FAT10 was phosphorylated upon IAV infection (Fig 4), we asked whether this modification could modulate its inhibitory function in response to RNA viruses. Accordingly, by using lentiviral transduction, we established three A549 cell lines stably expressing either FLAG-FAT10 wt, FLAG-FAT10 A, or FLAG-FAT10 E, respectively. Next, we infected these cells with IAV for 24 h and we measured the levels of IFN-$\beta$ in the supernatant by ELISA (Fig 6A). Interestingly, FLAG-FAT10 WT and FLAG-FAT10 E expression significantly reduced IFN-$\beta$ production, whereas FLAG-FAT10 A expression did not inhibit IFN-$\beta$ secretion as prominently as the WT or the phospho-mimicking form of FAT10 (FAT10 E) (Fig 6A). We further confirmed this finding in a kinetic experiment by showing the same trend also at 48 h after IAV infection (Fig 6B), suggesting that FAT10 phosphorylation leads to a gain of function in the FAT10-dependent reduction of the IFN-I secretion. To further verify this hypothesis, we studied an alternative model for RNA virus infection by using the vesicular stomatitis virus (VSV). VSV is a single-stranded, negative-sense RNA virus known to induce the RIG-I-mediated IFN-I response (Crill et al, 2015; Song et al, 2017; Munis et al, 2020). We transiently transfected the indicated variants of FLAG-FAT10 in A549 cells, followed by infection with VSV-GFP (GFP-tagged) for 24 h. Subsequently, we measured the secreted IFN-$\beta$ levels by ELISA. Also under these experimental conditions, FLAG-FAT10 A did not reduce IFN-$\beta$ secretion as efficiently as FLAG-FAT10 WT or the phospho-mimetic variant FLAG-FAT10 E (Fig 6C). Finally, we used VSV-GFP–infected cells to perform an immunoblot analysis. We observed that only FAT10 WT and FAT10 E, but not FAT10 A, strongly reduced the phosphorylation of the IFN-$\beta$ transcription factor IRF3 (Fig 6D, lanes

6 and 8), suggesting that FAT10 phosphorylation is crucial in the impairment of the IFN-I signaling pathway.

**Phosphorylation modulates the FAT10-mediated activation of OTUB1**

FAT10 was described to exert its inhibitory function by affecting the stability and activation of RIG-I (Nguyen et al, 2016; Wang et al, 2019). Therefore, we examined whether phosphorylated FAT10 could modulate IFN-I signaling by modulating RIG-I activity. To use similar conditions to the previous reports (Nguyen et al, 2016; Wang et al, 2019), we transfected the A549 cells described in Fig 6A with Poly (I:C) for 24 h, and performed an immunoblot analysis. We found in our experiments that the increased levels of endogenous RIG-I after Poly (I:C) transfection were not reduced, but even slightly enhanced in the presence of WT FAT10 (wt) or the FAT10 phospho-mutants (Fig 7A and B). Moreover, we did not observe any significant accumulation of RIG-I in the insoluble fraction upon FLAG-FAT10 overexpression (Fig S4A), or upon induction of endogenous FAT10 expression with TNF/IFNγ (Fig S4B) in IAV-infected A549 WT or FAT10 KO cells.

Non-covalent interaction of FAT10 with OTUB1 has been shown to increase OTUB1-mediated deubiquitylation of TRAF3 (Bialas et al, 2019). Therefore, we performed a FLAG immunoprecipitation combined with an immunoblot analysis to study the interaction of OTUB1 and WT FAT10 or the FAT10 phospho-mutants in HEK293 cells (Fig 7C). We observed that the FLAG-FAT10 A mutant interacted less efficiently with OTUB1 as compared with WT FLAG-FAT10 (wt) or the FLAG-FAT10 E mutant (Fig 7C, lanes 6–8). To assess whether this impaired interaction might influence the function of FAT10 in enhancing the enzymatic activity of OTUB1, we co-immunoprecipitated TRAF3 and HA-tagged ubiquitin to monitor the ubiquitylation status of the E3 ligase TRAF3 (Fig 7D). Over-expression of OTUB1 significantly reduced TRAF3 polyubiquitylation and co-expression of WT FAT10 (wt) further reduced this modification (Fig 7D, lanes 6 and 7), in line with our previous report (Bialas et al, 2019). Remarkably, co-expression of FAT10 E strongly diminished ubiquitylation of TRAF3 (Fig 7D, lane 9), whereas this effect could not be achieved by co-expression of FAT10 A (Fig 7D, lane 8). This result, which is in line with the experiment shown in Fig 7C, suggests that FAT10 phosphorylation is important for the OTUB1–FAT10 interaction. In addition, this modification contributes to the enhancement of OTUB1 activity in cleaving polyubiquitin from TRAF3. To further confirm that the negative effect that FAT10 has on the IFN-I response is mediated by binding of phosphorylated FAT10 to OTUB1, we

**Figure 5. FAT10 phospho-mutants do not differ from FAT10 WT.**
**(A)** A549 cells stably expressing WT, phosphorylation-deficient (A) or phospho-mimetic (E) versions of FLAG-FAT10 were stimulated with TNF, as indicated. 1 d later, cells lysates were prepared and subjected to a FLAG-immunoprecipitation and a Phos-tag/SDS–PAGE and immunoblot analysis with the indicated antibodies. **(B)** Quantification of the ECL signals from three independent experiments as shown in (A). Values were normalized to the respective FLAG-FAT10 (WT, -A, or -E) expression in the lysate (load). The value of WT FLAG-FAT10–expressing cells was set to 100% and all other values were calculated accordingly. **(C)** HEK293 cells were transiently transfected with constructs expressing the indicated forms of FLAG-FAT10. 1 d later, cells were lysed followed by FLAG-immunoprecipitation, SDS–PAGE, and immunoblot analysis with the indicated antibodies. **(D)** HEK293 cells were transiently transfected with constructs expressing the indicated forms of FLAG-FAT10 followed by a cycloheximide (CHX) chase over 5 h. Where indicated, cells were additionally treated for 6 h with MG132 (10 µM). The immunoblot was performed with the indicated antibodies. GAPDH was used as loading control. **(E)** Quantification of the ECL signals from three independent experiments as shown in (D). Levels were normalized to the respective levels of the housekeeping gene GAPDH. Values at 0 h were set to unity and the other values were calculated accordingly. **(F)** A549 cells were transiently transfected with constructs expressing the specified forms of FLAG-FAT10 (green) and the cellular localization of FAT10 was assessed by confocal microscopy. Nuclei were stained with DAPI. Scale bars represent 20 µm. One representative example out of three independent experiments is shown.
Source data are available for this figure.

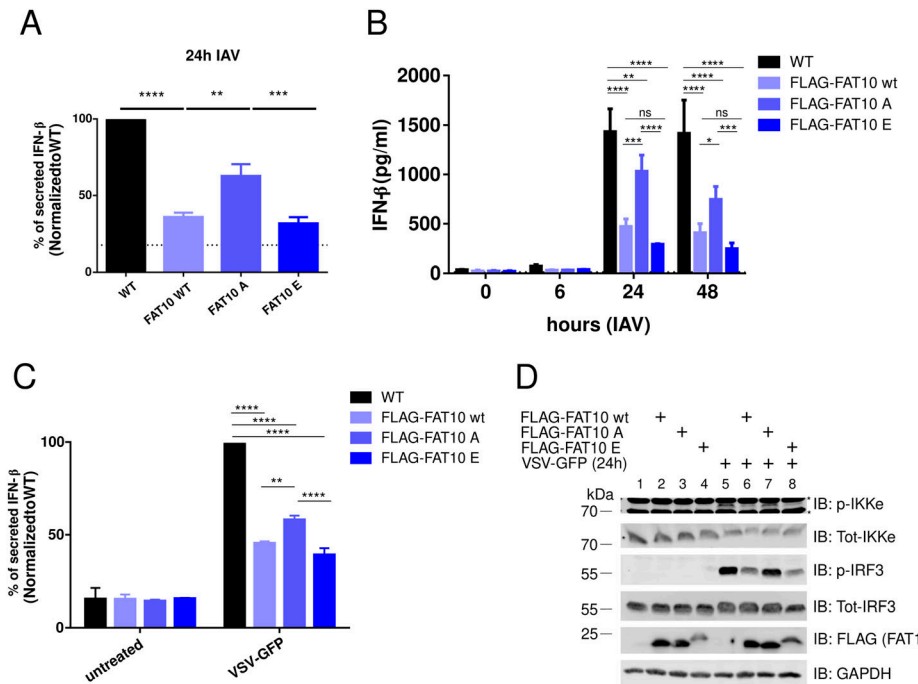

**Figure 6. The phospho-mimetic FAT10 mutant diminishes IFN-I secretion.**
**(A)** A549 WT and stably transfected A549 FLAG-FAT10, A549 FLAG-FAT10 A, and A549 FLAG-FAT10 E cell lines were infected with influenza A virus (MOI:1). After 24 h, the supernatants were collected and an IFN-β ELISA was performed. **(B)** The experiment was performed as in (A) and the supernatants were collected 0, 6, 24, and 48 h after influenza A virus infection. **(C)** A549 cells were transiently transfected twice (at 0 and 24 h) with the indicated FLAG-FAT10 expression plasmids. After 48 h, cells were infected with vesicular stomatitis virus-GFP (MOI:0.1) for 1 h. After 24 h, the supernatants were collected and the concentration of IFN-β in the supernatant was measured by ELISA. **(D)** The cell pellets from (C) were lysed and protein expression was analyzed by immunoblotting using the antibodies indicated. Data in (A, B, C) are mean ± SEM (n = 3); P < 0.05 (t test); ns: not significant; Asterisks in (D) mark unspecific background signals.
Source data are available for this figure.

measured the secretion of IFN-β in A549 wt and A549 OTUB1-deficient cells in the presence of WT FAT10 or phospho-mimetic FAT10 (FAT10 E). We found that the Poly (I:C)-induced IFN-β production (Fig 7E) and the IRF3 phosphorylation (Fig S5A) was significantly reduced by WT FLAG-FAT10 (wt), and that this negative response was further reduced upon FLAG-FAT10 E expression (Figs 7E and S5A). Remarkably, this effect was abolished in OTUB1 KO cells (Figs 7E and S5A), which strongly suggests that the activation of OTUB1 by phosphorylated FAT10 is responsible for the FAT10-mediated impairment of the antiviral IFN-I response. To strengthen these data, the experiment was repeated, however, instead of transfection of the cells with Poly (I:C), A549 WT, and OTUB1 KO cells were infected with IAV for 24 h (Figs 7F and S5B). Confirming the results obtained upon Poly (I:C) transfection in Fig 7E, FLAG-FAT10 E showed also here the strongest negative impact on IFN-β secretion, whereas again in OTUB1 KO cells, this inhibitory effect was abolished (Figs 7F and S5B). As a final proof, the experiments were repeated under completely endogenous conditions and IFN-β secretion was monitored in A549 WT, FAT10 KO, OTUB1 KO, and FAT10/OTUB1 double KO cells (Fig S6A). The different cell lines were either transfected with Poly (I:C) (Fig 7G), or infected with IAV (Fig 7H), and additionally treated with TNF/IFNγ to induce the expression of endogenous FAT10. Under both experimental conditions, induction of endogenous FAT10 expression significantly diminished IFN-β secretion in A549 WT cells, which was not observed in FAT10 knockout cells, again confirming that this effect is mediated by FAT10. Furthermore, a knockout of OTUB1 or of OTUB1 and FAT10 at the same time (FAT10 KO/OTUB1 KO) likewise abrogated the inhibitory effect of endogenous FAT10 induction on IFN-β secretion (Fig 7G and H). Of note, A549 OTIUB1 KO cells, already secreted low levels of IFN-β upon treatment with TNF/IFNγ, and even without IAV infection (Figs 7H and S6B), pointing to an alternative pathway for IFN-I induction, mediated by TNF/IFNγ in the absence of OTUB1.

In summary, our data show that FAT10 phosphorylation is induced either by TNF treatment or by IAV infection. Phosphorylation of FAT10 strengthens its interaction with OTUB1, thereby stabilizing OTUB1 and enhancing the OTUB1 deubiquitylase activity towards TRAF3, leading to down-regulation of the antiviral IFN-I response (Fig 8).

## Discussion

Earlier reports have described that ubiquitin and the ULM SUMO can be phosphorylated (Matic et al, 2008; Kane et al, 2014; Kazlauskaite et al, 2014; Koyano et al, 2014; Lazarou et al, 2015), shedding light on a complex network of modifications in which the "modifiers are modified" to create multiple states of a protein from a single gene product. In this study, we applied a phospho-proteomic approach through which we identified five phosphorylation sites of FAT10 (Fig 1, Table S1). Interestingly, the phosphorylation sites identified in FAT10 do not structurally overlap with the ones identified in ubiquitin (Swatek & Komander, 2016). This is in line with our previous report which shows that ubiquitin and FAT10 possess completely distinct surface charges (Aichem et al, 2018). Moreover, this finding is further supported by the observation that in vitro PINK1 is able to recognize and phosphorylate ubiquitin but not FAT10 (Fig 1C) (Kane et al, 2014). Vice versa, the kinase MK3 phosphorylated FAT10 but not ubiquitin, confirming that ubiquitin and FAT10 are two proteasome-targeting proteins with distinct protein-binding characteristics. Although we did not observe a MK3-mediated FAT10 phosphorylation in cells (Fig 2), we observed that FAT10 phosphorylation was strongly induced by TNF (Fig 2) and upon IAV infection (Fig 4). Moreover, using a "by exclusion"

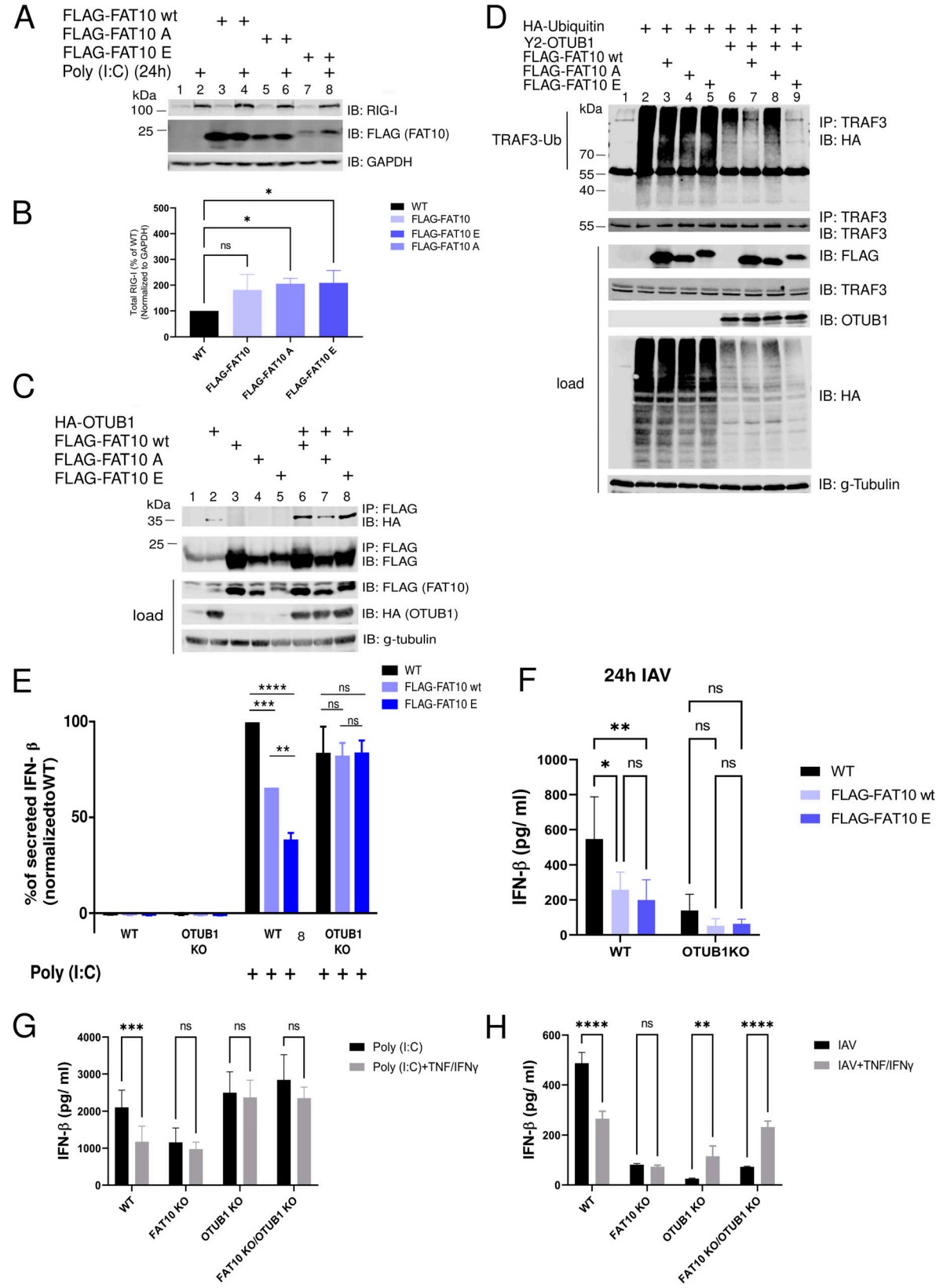

approach, we found that IKKβ is the major kinase for FAT10 phosphorylation in HEK293 and A549 cells (Figs 3 and 4). Nevertheless, we do not want to exclude that the two kinases JNK3 and IKKε might be able to phosphorylate FAT10 in different contexts, as for example in different tissues, because they were both able to modify FAT10 under in vitro conditions and upon their overexpression in HEK293 cells (Fig 3). Interestingly, in contrast to previous reports (Zhang et al, 2016; Wang et al, 2019), we found that FAT10 expression was not induced upon IAV infection, whereas it was strongly up-regulated by TNF and IFNγ, which both are produced by leukocytes during an antiviral immune response.

To elucidate the functional consequences of FAT10 phosphorylation, we performed a phospho-mimetic analysis (Fig 5), because this approach was informative in previous works on ubiquitin phosphorylation (Kane et al, 2014; Swaney et al, 2015). As a proof for the functionality of the mutated FAT10 variants, we found that the phospho-mutations neither significantly altered bulk FAT10 conjugation to substrate proteins, nor the speed of proteasomal degradation, nor its subcellular localization (Fig 5). These results indicated that the mutation of the phosphorylation sites did neither impact the key biochemical features of FAT10 nor did they change the overall stability of FAT10. Nonetheless, we found that the phospho-mimicking (E) mutation of FAT10 caused a gain of function in the inhibition of the IFN-I response upon IAV and VSV-GFP infections (Fig 6). Both investigated RNA viruses, IAV and VSV, activate the RIG-I/MAVS signaling platform, and this event leads to the recruitment of TRAF6, causing IKK-β activation (Yoboua et al, 2010; Liu et al, 2013a). Our data reveal that IKK-β does not only phosphorylate p65 and IκB during viral infection (Chen et al, 1996; Yoboua et al, 2010; Liu et al, 2013a), but show that IKK-β also phosphorylates FAT10. Thereby, phospho-FAT10 contributes to the inhibition of the IFN-I cascade in a negative-feedback fashion (Fig 7).

By investigating how FAT10 mechanistically acts as an inhibitor of IFN-β secretion, we found that the levels of activated RIG-I were only slightly enhanced upon the overexpression of any of the FAT10 variants (Fig 7A). Moreover, in contrast to a previous report (Nguyen et al, 2016) we found that neither overexpressed FLAG-tagged, nor TNF/IFNγ induced endogenous FAT10 did sequester RIG-I into the

insoluble fraction during an IAV infection (Fig S4). Instead, we discovered that phosphorylation-deficient FAT10 (FAT10 A) partially lost the capacity of binding to OTUB1 and to reduce the OTUB1-dependent deubiquitylation of TRAF3 (Fig 7). TRAF3 is a crucial scaffold molecule that, once auto-ubiquitylated, recruits the TBK1/IKKε kinase complex leading to IRF3 phosphorylation (Li et al, 2010; Häcker et al, 2011). Based on these results, we suggest that the IKKβ-mediated phosphorylation of FAT10 serves as a negative modulator of excessive IFN-I secretion during viral infection, which likely contributes to an increased viral replication (Samuel, 2001). Moreover, our work indicates that this effect could be directly mediated by the binding of phospho-FAT10 to OTUB1, which activates this DUB leading to enhanced deubiquitylation of TRAF3. On the other hand, a recent report described that OTUB1 localizes to mitochondria during IAV infection, where it decreases the levels of K48-linked ubiquitin chains on RIG-I, thus counteracting its proteasomal degradation (Jahan et al, 2020). Hence, it would be crucial to investigate the ubiquitylation status of RIG-I in the presence of FAT10 and phospho-FAT10. This may help to understand whether phospho-FAT10, while inducing the deubiquitylation of TRAF3, could simultaneously enhance OTUB1 activity in removing K48-ubiquitin chains from RIG-I, thereby impairing its proteasomal degradation.

It is interesting to note that the measured IFNβ levels differed always markedly between A549 OTUB1 knockout cells which were treated either with Poly (I:C), or which were infected with IAV (Fig 7E and F). Although we always observed a clear decrease in IFNβ secretion upon infection with IAV in A549 OTUB1-KO cells as compared with WT cells (black bars in Fig 7F), the secreted IFNβ levels remained high in OTUB1 knockout cells which were transfected with Poly (I:C) (black bars in Fig 7E). This might be explained by the robustness of RIG-I activation caused by differences because of a transfection of Poly (I:C) as compared with RIG-I activation upon a viral infection. This might eventually result in the activation of other signaling pathways, contributing to IFNβ secretion. Of course, the exact mechanism causing this difference should be investigated in future experiments.

Another open question concerns the dynamic of FAT10 phosphorylation in vivo. FAT10 is overexpressed during lymphocytic choriomeningitis virus infection in mice, where it counteracts IFN-β

---

**Figure 7. Phosphorylated FAT10 activates OTUB1 deubiquitylase activity.**
**(A)** A549 WT and stably transduced A549 FLAG-FAT10, A549 FLAG-FAT10 A and A549 FLAG-FAT10 E expressing cells were lysed 24 h after Poly (I:C) transfection. The lysates were subjected to SDS–PAGE and immunoblot analysis with the antibodies indicated. **(B)** Quantification of the RIG-I expression levels calculated from the immunoblot shown in (A), normalized to GAPDH. **(C)** HEK293 cells were transiently transfected with expression plasmids for HA-OTUB1, FLAG-FAT10 WT, -A, and -E variants, as indicated. After 24 h, cells were harvested and lysed. Subsequently, an immunoprecipitation against the FLAG tag was performed, followed by SDS–PAGE and immunoblot analysis with the antibodies indicated. γ-tubulin served as loading control. **(D)** HEK293 cells were transiently transfected with expression plasmids for HA-ubiquitin, Y2-OTUB1, and the different FLAG-FAT10 variants, as indicated. After 24 h, cells were harvested and lysed. Subsequently, endogenous TRAF3 was immunoprecipitated with a monoclonal TRAF3-reactive antibody followed by SDS–PAGE and immunoblot analysis with the indicated antibodies. γ-tubulin served as loading control. **(E)** Non-transfected A549 WT and A549 OTUB1 KO cells (WT) (black bars), and transiently with FLAG-FAT10 wt (light blue) or FLAG-FAT10 E (dark blue) expression plasmids transduced A549 and A549 OTUB1 KO cells were additionally transfected with Poly (I:C) for 24 h. The supernatant was subjected to IFN-β ELISA. Levels were normalized to the levels of the respective non-transfected sample. Data are mean ± SEM (n = 3). **(F)** A549 WT and OTUB1 KO cells were left untreated (black bars) or were transiently transduced with expression plasmids for FLAG-FAT10 WT (light blue) or the FLAG-FAT10 E mutant (dark blue). After 48 h, cells were infected with Influenza A virus (MOI:1) for 1 h. After 24 h, the supernatants were collected and the concentration of IFN-β in the supernatants was measured by ELISA. **(G)** Endogenous FAT10 expression was induced by treating A549 WT, FAT10 KO, OTUB1 KO, and FAT10 KO/OTUB1 KO cells with TNF/IFNγ for 24 h. Then, cells were transfected with Poly (I:C) and stimulated at the same time with TNF/IFNγ for the next 24 h. The supernatants were collected and the concentration of IFN-β in the supernatants was measured by ELISA. Data are mean ± SEM (n = 3). **(H)** Endogenous FAT10 expression was induced by treating A549 WT, FAT10 knockout (KO), OTUB1 KO, and FAT10 KO/OTUB1 KO cells with TNF/IFNγ for 24 h. Subsequently, cells were infected with influenza A virus (MOI:1) for 1 h. Cells were again stimulated with TNF/IFNγ for the next 24 h after which the supernatant was collected and the concentration of IFN-β in the supernatant was measured by ELISA. Data are mean ± SEM (n = 5). P < 0.05 (t test), ns: not significant. Source data are available for this figure.

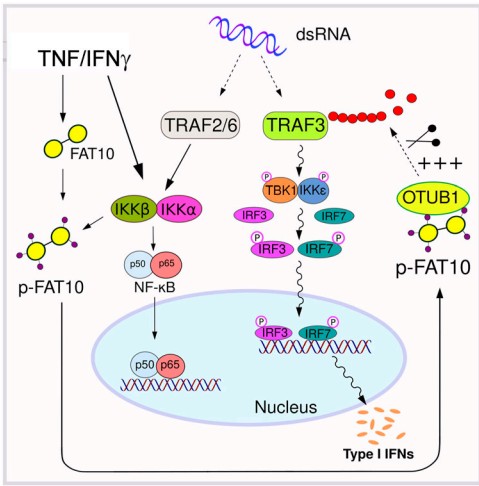

**Figure 8. Cartoon summarizing how FAT10 down-regulates IFN-I secretion.**
FAT10 is phosphorylated upon TNF treatment or influenza A virus infection mainly by IKK$\beta$. Phosphorylated FAT10 binds stronger to the deubiquitinating enzyme OTUB1 causing its stabilization. As a consequence, TRAF3 is deubiquitylated and inactivated, ultimately leading to reduction of interferon regulatory factor 3 phosphorylation, which causes a down-regulation of the antiviral IFN-I response.

secretion and increases IFN$\gamma$-induced inflammation (Mah et al, 2019). Thus, based on the fact that we could not observe a virus-induced induction of FAT10 expression in A549 cells (Fig 4), we speculate that the inflammatory microenvironment that accompanies the viral infection in vivo (which includes TNF and IFN$\gamma$) stimulates FAT10 expression (Harty et al, 2000). Subsequently, both TNF and the viral infection itself may synergistically stimulate the IKK$\beta$-mediated phosphorylation of FAT10, with a mitigating outcome on IFN-I secretion.

In summary, we report that under inflammatory conditions, FAT10 is expressed and is phosphorylated by IKK$\beta$ in response to TNF and to viral infection. Thereby, FAT10 phosphorylation strengthens the FAT10–OTUB1 interaction, causing a reduced polyubiquitylation of TRAF3 which strongly correlates with the reduced antiviral IFN-I secretion caused by phospho-FAT10. Accordingly, we discovered FAT10 and its phosphorylation as putatively aggravating factors during RNA viral infections. As a conclusion, an inhibition of FAT10 expression and/or FAT10 phosphorylation, as for example by IKK$\beta$ inhibitors, might provide a potential druggable target to ameliorate the response to RNA viruses.

# Materials and Methods

## Cell culture and cell lines

HEK293T, HEK293, A549, A549 FLAG-FAT10 WT, A549 FLAG-FAT10 Ser62A Ser64A Thr77A Ser95A Ser109A (A549 FLAG-FAT10 A), A549 Ser62E Ser64E Thr77D Ser95E Ser109E (A549 FLAG-FAT10 E), A549 OTUB1 KO, A549 OTUB1 KO FLAG-FAT10 Ser62A Ser64A Thr77A Ser95A Ser109A (A549 OTUB1 KO FLAG-FAT10 A), A549 OTUB1 KO Ser62E Ser64E Thr77D Ser95E Ser109E (A549 OTUB1 KO FLAG-FAT10 E), A549

FAT10 KO, and A549 FAT10 KO/OTUB1 KO cells were cultivated in DMEM (Thermo Fisher Scientific) supplemented with 10% FCS (Gibco/Thermo Fisher Scientific), 1% stable glutamine (100x, 200 mM), and 1% penicillin/streptomycin (100x) (both from Biowest/VWR).

## DNA constructs

The cDNA construct encoding His-3xFLAG-FAT10 (FLAG-FAT10) wt (pcDNA3.1) was described in (Chiu et al, 2007). FLAG-FAT10 Ser62A, Ser64A, Thr77A, Ser95A, Ser109A (FLAG-FAT10 A), and FLAG-FAT10 Ser62E Ser64E, Thr77D, Ser94E, Ser109E (FLAG-FAT10 E) were generated by site-directed mutagenesis of the FLAG-FAT10 wt construct using the QuickChange II Site-directed mutagenesis kit (200521; Agilent Technologies). The constructs for lentiviral transduction of the FLAG-FAT10 wt, FLAG-FAT10 E, and FLAG-FAT10 A genes were obtained by direct cloning of the FAT10 inserts from the above described pcDNA3.1 plasmids into a pCDH vector using the restriction enzymes Nhe I and Not I (NEB). The cDNA constructs encoding HA-MK3 wt, HA-MK3 TT/EE, and HA-MK3 TT/AA have been described in Ludwig et al (1996) and were a kind gift of the Gaestel group (Institute for Cell-Biochemistry, Hannover). The following cDNA constructs were obtained from Addgene: FLAG-JNK1 (plasmid #13798, [Dérijard et al, 1994]), JNK2 (plasmid #13755, [Gupta et al, 1996]), JNK3 (plasmid #13758, [Gupta et al, 1996]), HA-IKK$\alpha$ (plasmid #15469, [Nakano et al, 1998]), FLAG-IKK$\beta$ (plasmid #15470, [Nakano et al, 1998]), HA-IKK$\gamma$ (plasmid #13512, [Tang et al, 1999]), FLAG-IKK$\varepsilon$ (plasmid #26201, [Fitzgerald et al, 2003]), pM2.G (plasmid #12259; kind gift from Didier Trono), and psPAX2 (plasmid #12260; kind gift from Didier Trono). The HA-OTUB1 and Y2-OTUB1 plasmids are published in Bialas et al, 2019.

## Antibodies

Antibodies used for immunoblotting are as follows: mouse anti-FLAG antibody (F1804, 1:3,000; Merck), mouse anti-FLAG (HRP) antibody (A8592, 1:3,000; Merck), rabbit anti-FLAG antibody (F7425, 1:750; Merck), mouse anti-HA antibody (H3663, 1:5,000; Merck), rabbit anti-HA antibody (H608, 1:1,000; Merck), rabbit anti-GAPDH antibody (G9545, 1:10,000; Merck), mouse anti-tubulin antibody (T6557, 1:10,000; Merck), rabbit anti-FAT10 antibody ([Roverato et al, 2021] 1:750), rabbit anti-MK3 antibody (3043, 1:1,000; Cell Signaling), rabbit anti-IKK$\beta$ antibody (2684, 1:1,000), rabbit anti-IRF3 antibody (11904, 1:1,000; Cell Signaling), rabbit anti-phospho-IRF3 antibody (4947, 1:1,000; Cell Signaling), rabbit anti-IKK$\varepsilon$ (3416, 1:1,000; Cell Signaling), rabbit anti-phospho-IKK$\varepsilon$ antibody (06-1340, 1:1,000; Merck), mouse anti-RIG-I antibody (MABF297, 1:1,000; Merck), mouse anti-OTUB1 antibody (CF505157, 1:1,000; Thermo Fisher Scientific), mouse anti-M1 antibody (ab22395, 1:1,000; Abcam), rabbit anti-TRAF3 antibody (ab239357, 1:1,000; Abcam), 800CW goat anti-mouse IgG (926-332210 1:10,000; Licor), and 680RD goat anti-rabbit antibody (926-68071, 1:10,000; Licor). Antibodies for immunoprecipitations were used as follows: mouse anti-FLAG antibody (F1804; Merck), rabbit anti-phospho-serine antibody (ab9332; Abcam), mouse anti-FAT10 antibody (clone 4F1, Enzo Life Science [Aichem et al, 2010]), mouse anti-HA antibody (H3663; Merck), and mouse anti-TRAF3 antibody (sc-6933; Santa Cruz). Mouse anti-FLAG antibody (F1804, 1:1,000;

Merck) and goat anti-mouse Alexa Fluor 568 (A-11004, 1:1,000; Thermo Fisher Scientific) were used for immunofluorescence.

## Virus strains and viral infection

The IAV strain A/Regensburg/D6/09 (H1N1pdm09; RB1) was a kind gift of Oliver Planz, Tuebingen University, Germany, and was produced as described by Mah and colleagues (Mah et al, 2019). The VSV-GFP strain Indiana virus was a kind gift from Daniel Pinschewer, Basel University, Switzerland. In all the experiments involving IAV and VSV-GFP, the cells were incubated for 1 h with IAV (MOI: 1) or VSV-GFP (MOI: 0.1) in serum-free DMEM medium, followed by the replacement of the infection medium with standard medium. Cells were incubated for 24 h before supernatant collection and/or cell lysis.

## In vitro kinase assay

In the in vitro phosphorylation assay, the recombinant kinase was mixed together with its recombinant substrate protein in required concentrations and resuspended in the kinase buffer (20 mM Tris–HCl, 10 mM MgCl$_2$, 0.1 mM EDTA, 2 mM DTT 0.01%, 4 mM ATP) in a total volume of 10 $\mu$l. The kinase reaction took place at 30°C with shaking for 45 min using Eppendorf Thermomixer Comfort (Eppendorf). Afterwards, samples were mixed with 10 $\mu$l 4x sample buffer and stored at 20°C or subsequently used for the following analysis. In this work, the following recombinant kinases were used: 1 $\mu$g PINK1 (AP-180-100; Novus Bio), 1 $\mu$g MK3 (0633-0000-1; Reaction Biology), 1.5 $\mu$g GST-His-IKK$\beta$ (0258-0000-1; Reaction Biology), and 1.5 $\mu$g GST-His-IKK$\varepsilon$ (0320-0000-1; Reaction Biology). The kinases were incubated with 4 $\mu$g of recombinant ubiquitin or with 1.5 $\mu$g of recombinant FAT10 (Aichem et al, 2019), as indicated.

## Radiolabeled samples

1 $\mu$g recombinant PINK1 (AP-180-100; R&D) and 1 $\mu$g recombinant MK3 (0633-0000-1; Reaction Biology) were mixed with 4 $\mu$g recombinant ubiquitin or with 1.5 $\mu$g recombinant FAT10, as indicated. The proteins were incubated in 10 $\mu$l kinase buffer (20 mM Tris–HCl, 10 mM MgCl, 0.1 mM EDTA, 2 mM DTT 0.01%, 4 mM ATP) in presence of 1 $\mu$Ci $\gamma$-[$^{32}$P] ATP at 30°C for 15 or 45 min. After that, 4x SDS–PAGE sample buffer was added and proteins were separated by SDS–PAGE. Incorporation of radioactive ATP was measured by autoradiography.

## IP-kinase assay

Immunoprecipitated kinases were exposed to recombinant FAT10 in an in vitro phosphorylation reaction. After overexpression of the indicated kinases in HEK293 cells, an immunoprecipitation was performed using antibodies reactive against the respective tag (FLAG or HA). Afterwards, beads were extensively washed with buffers NET-TN (50 mM Tris HCl pH 8.0, 650 mM NaCl, 0.5% Triton X-100) and NET-T (50 mM Tris HCl pH 8.0, 150 mM NaCl, pH 7.8, 0.5% Triton X-100). Immediately after the washing steps, the immunoprecipitated proteins were incubated with 1.5 $\mu$g recombinant FAT10 in kinase buffer (see above) for 45 min at 30°C with gently shaking.

The reaction was terminated by the addition of 10 $\mu$l 4x sample buffer and samples were stored at –20°C or directly analyzed by Phos-tag SDS–PAGE.

## Radiometric protein kinase filter-binding assay

The radiometric protein kinase filter-binding assay was performed by Proqinase. Further information can be found at https://www.reactionbiology.com/.

## Phosphopeptide enrichment and LC-MS

Endogenous or tagged FAT10 affinity-purified samples were separated by SDS–PAGE and Coomassie-stained bands (corresponding to FAT10 protein) were cut out. Proteins were reduced, alkylated, and digested in-gel with the endoproteinase trypsin. Peptides were extracted and phosphopeptides were enriched by TiO$_2$ chromatography as described in Oellerich et al (2009). In brief, peptides were dissolved with 20 $\mu$l of 200 mg 2,5-dihydroxybenzoic acid (Sigma-Aldrich) in 80% acetonitrile (ACN), 5% trifluoroacetic acid and loaded onto a TiO$_2$ column. The column was washed three times with 20 $\mu$l of 200 mg 2,5-dihydroxybenzoic acid in 80% ACN, 5% trifluoroacetic acid and five times with 20 $\mu$l of 80% ACN, 5% trifluoroacetic acid. The column was then incubated three times with 20 $\mu$l of 0.3 normal (N) NH$_4$OH, pH ≥ 10.5, to elute phosphopeptides. LC-MS analysis of phosphopeptides was performed under standard conditions on an Orbitrap HF instrument (Thermo Fisher Scientific) working in data-dependent acquisition mode using top 15 method. MS was coupled to an UltiMate LC system (Thermo Fisher Scientific) equipped with C18 trap column packed in-house (1.5 cm, 360-$\mu$m outer diameter, 150-$\mu$m inner diameter, Nucleosil 100-5 C18; MACHEREY-NAGEL, GmbH & Co. KG), and an analytical C18 capillary self-made column (30 cm, 360-$\mu$m outer diameter, 75-$\mu$m inner diameter, Nucleosil 100-5 C18). The flow rate was 300 nl/min, with a gradient from 8% to 40% ACN in 0.1% (vol/vol) formic acid for 40 min. Data were processed with MaxQuant software (version 1.5.2.8) and searched against the Uniprot database (taxonomy human, 22.01.2014, 155143 entries) with phosphorylation at serine, threonine, and tyrosine residues and methionine oxidation, and cysteine carboxyamidomethylation as variable modifications. The mass spectrometry proteomics data have been deposited to the ProteomeXchange Consortium (Deutsch et al, 2023) via the PRIDE (Perez-Riverol et al, 2019) partner repository with the dataset identifier PXD028918.

## Transient transfection of plasmids

HEK293 or A549 cells were seeded and cultured until 60–80% confluency was reached. For transfection of HEK293 cells, PEI transfection reagent (408727; Merck), and for transfection of A549 cells, FuGENE transfection reagent (E2311; Promega) was used. In case of a 100-mm$^2$ cell culture dish, 600 $\mu$l serum and antibiotic-free medium was mixed with 18 $\mu$l transfection reagent and left for 5 min at RT. 6 $\mu$g of plasmid DNA was added, mixed, and after additional 20 min of incubation at RT, the suspension was added dropwise to the cells. For dishes/well plates of other sizes, the amounts of reagents were adapted. Cells were transfected with

equal amounts of plasmid DNA and were co-transfected with the same amount of overall plasmid DNA. When required, empty pcDNA3.1-His/-A (V38520; Invitrogen) was used to balance plasmid amounts. For transfection of low molecular weight Poly (I:C) (tlrl-picw-250; Invivogen), 1 μl/ml of Poly (I:C) was transfected with FuGENE transfection reagent (ratio Poly [I:C]: FuGENE 1:6).

### Cell extracts, immunoprecipitation, starvation/TPA treatment, and CHX experiments

As a standard preparation of cellular samples, cells were harvested using trypsin and centrifuged for 7 min at 1,000$g$. The obtained cell pellet was lysed in Triton-X 100 lysis buffer (20 mM Tris–HCL pH 7.6, 50 mM NaCl, 10 mM MgCl$_2$, 1% Triton X-100, 1x protease inhibitor mix, and 1x phosphatase inhibitor cocktail) and incubated for 20 min at 4°C, followed by centrifugation (10 min, 13,000$g$, 4°C). Samples subjected to phosphorylation analysis were lysed in lysis buffer containing an additional 1x phosphatase inhibitor (PhosSTOP, Roche). The lysate was analyzed by SDS–PAGE/immunoblot and/or subjected to immunoprecipitation. Immunoprecipitations were performed in the case of FLAG-tagged proteins with 25 μl of EZview Red ANTI-FLAG M2 Affinity Gel (F2426; Merck), and in case of HA-tagged proteins with EZview Red Anti-HA Affinity Gel (E6779). 30 μl of EZview Red Protein A Affinity Gel (P6486; Merck) and 5 μg of the monoclonal FAT10-reactive antibody 4F1 ([Aichem et al, 2010] and Enzo Lifesciences) were used for immunoprecipitation of endogenous FAT10. After incubation for 4 h at 8°C, beads were washed twice with NET-TN buffer and subsequently twice with NET-T buffer, followed by addition of 4x Laemmli sample buffer and SDS–PAGE/immunoblot analysis. To activate the PKC kinase, cells were starved for 24 h (0.3% FCS DMEM) followed by TPA (P8139; Sigma-Aldrich) treatment (30 min) before lysis.

For the CHX chase experiments, cells were treated with CHX (final concentration 10 μg/ml in DMSO) for the indicated time points before lysis. Where indicated, cells were simultaneously treated with 5 μM MG132.

### SDS–PAGE and immunoblotting

Sodium dodecyl sulfate polyacrylamide gel-electrophoreses (SDS–PAGE) was performed to separate proteins according to the molecular mass. Before loading onto the gels, samples were boiled in SDS sample buffer (250 mM Tris–HCl pH 6.8, 40% glycerol, 8% SDS, 0.004% Bromophenol blue) at 95°C for 5 min. Along with 5 μl PageRuler pre-stained protein ladder (26617; Thermo Fisher Scientific) as a marker, the samples were loaded onto a 1.5 mm SDS–PAGE gel. The electrophoretic separation was performed in 1x SDS–PAGE running buffer (25 mM Tris–HCl pH 8.3, 0.192 M glycine, 0.1% SDS). Proteins were blotted onto 0.45 μm nitrocellulose membranes using the criterion wet-blotter (Bio-Rad) according to manufacturer's instructions. The protein transfer was conducted in 1x SDS–PAGE blotting buffer (25 mM Tris–HCl pH 8.3, 0.192 M glycine, 20% methanol) at a constant voltage of 110 V for 90 min. Blotted membranes were blocked in Odyssey Blocking Buffer (LI-COR) for 1 h at RT. Blocked membranes were incubated with the primary antibody diluted in Odyssey Blocking Buffer (LI-COR) at needed concentrations overnight at 4°C. Then, the blots were washed three times with TBS-T (20 mM

TBS, 137 mM NaCl, Tween-20 0.2%) for 5 min each and incubated with the respective secondary antibody at needed concentrations for 2 h at RT. Blots were washed again for three times in TBS-T buffer before the protein bands could be detected using LI-COR Odyssey Fc Imager (LI-COR) and Image Studio Lite software (LI-COR). The times for developing varied between 90 and 180 s.

### Phos-tag–based mobility shift detection of phosphorylated proteins

One of the following methods was used to detect phosphorylation of FAT10 by using Phos-tag SDS–PAGE. These specific SDS gels can be used for the simultaneous analysis of a phospho-protein isoform and its non-phosphorylated counterpart. The phosphorylated isoforms can be separated depending on the site or number of the phosphate groups. In this study, the Zn$^{2+}$-Phos-tag SDS–PAGE was applied according to the manufacturer's instructions (AAL-107M; Labchem-Wako). The gel was run with 1x Phos-tag SDS–PAGE running buffer (Tris base 0.5 M, MOPS 0.5 M, SDS 0.5%), for 20 min at 60 V until the proteins entered the stacking gel, followed by separation at 110 V. SuperSep Phos-tag gels (cat. #195–17991; WAKO Chemical) were also used to separate samples following the manufacturer's instructions. Immediately after the SDS–PAGE, the gels were soaked in 1 mM EDTA for 10 min (Zn$^{2+}$-Phos-tag SDS–PAGE) or in 10 mM EDTA for 30 min (SuperSep Phos-tag gel) with gentle agitation. Then, the gels were soaked in SDS–PAGE blotting buffer (25 mM Tris pH 8.3, 192 mM glycine, 20% methanol) for 10 min with gentle agitation. Blotting was performed using PVDF (which were incubated in 100% methanol for 5 min at RT before usage) or nitrocellulose membranes. The following steps were the same as in immunoblot analysis (see above).

We also used tris-tricine protein/peptide separation gels (https://depts.washington.edu/bakerpg/protocols/tris_tricine_gels.html) in combination with Zn$^{2+}$-Phos-tag SDS–PAGE (AAL-107M; Labchem-Wako). Briefly, 29:1 acrylamide/bisacrylamide, Tris HCl/SDS pH 8.45, Phos-tag AAL-107, H$_2$O, Glycerol, 10% (wt/vol) ammonium persulfate and TEMED were used to cast separating and stacking gel (AAL-107; without Phos-tag). The gel was run with 1x Phos-tag SDS–PAGE running buffer (Tris base 0.5 M, MOPS 0.5 M, SDS 0.5%) for 20 min at 60 V until the proteins entered the stacking gel followed by separation at 110 V. To show that the bands are specific for phosphorylated proteins, same samples were also run in the gels without the addition of Phos-tag AAL-107 in a cathode buffer (0.1 M Tris, 0.1 M Tricine, and 0.1% SDS) and an anode buffer (0.2 M Tris–HCl, pH 8.9) at 10 mA for 35 min and at 20 mA for 1 h 30 min. Blotting was performed using a standard semi-dry blotting technique on nitrocellulose membrane. The following steps were the same as in immunoblot analysis (see above).

### Triton X-100-based soluble/insoluble fractionation

To assess the presence of the FAT10 and RIG-I in the soluble versus the insoluble fraction, cells were harvested and lysed as described above. Cells were subsequently centrifuged at 14,000$g$ for 15 min. The supernatant (soluble fraction) was harvested and the pellet (insoluble fraction) was washed with PBS and solubilized with a SDS-based lysis buffer (20 mM Tris–HCl pH 7.8, 1 mM MgCl$_2$, 1%

Triton-X100, 4% SDS, 10% glycerol, protease inhibitor, phosphatase inhibitor). The soluble and insoluble samples were subjected to immunoprecipitation and immunoblot analysis.

### ELISA assay

A549 cells were seeded in a 6-well plate and infected with IAV strain A/Regensburg/D6/09 (H1N1pdm09; RB1) at a MOI of 1 or with VSV-GFP Indiana (MOI: 0.1) for 24 h. A classical sandwich-ELISA for IFN-$\beta$ was performed with the supernatants of the stimulated cells, according to the manufacturer's protocol (41410; R&D System). Briefly, anti-IFN-$\beta$ antibody was pre-coated onto 96-well plates. The plates were washed two times with washing buffer (PBS, 0.2% Tween-20, 200 $\mu$l/well). 100-$\mu$l standards and samples in duplicates were added to each well and incubated for 90 min at 37°C, followed by two washing steps. Subsequently, 100 $\mu$l of the biotin-labeled antibody was added into each well and further incubated for 60 min at 37°C, and washed two times. Finally, wells were incubated with 100 $\mu$l HRP-streptavidin conjugate (SABC) for 30 min, washed five times, and incubated with 90 $\mu$l/well TMB substrate for 30 min to visualize the HRP enzymatic reaction displayed by blue color change in the samples. The reaction was stopped by adding 50 $\mu$l of STOP solution. The absorbance was measured at 450 nm in a TECAN Infinite M200 Pro plate reader and the concentration of IFN-$\beta$ was calculated. All the experiments were repeated at least three times and data shown are means ± SEM.

### Real-time PCR

Real-time PCR was performed as previously described (Schregle et al, 2018). In brief, RNA was purified using the RNeasy Plus Mini Kit (QIAGEN) according to the manufacturer's instructions. For synthesis of single-stranded cDNA from total RNA, the Reverse Transcription System (Promega) was used. Relative gene expression was measured using the LightCycler Fast Start DNA Master SYBR Green I kit (Roche). The 96 Real-Time PCR-Thermocycler and the AccuSEQ Real-Time PCR Software v3.1 (both from Analytik) was used. The primers to measure FAT10 expression were as follows: forward 5'-GGGATTGACAAGGAAACCACTA-3'; reverse 5'-TTCACAACCTGCTTCTTAGGG-3'. The primers for Rpl13a (control) were as follows: forward 5'-CTACAGAAACAAGTTGAAGTACCTG-3'; reverse 5'-ATGCCGTCAAACACCTTGAG-3'.

### Production of lentiviral particles

HEK293T cells were transfected with pMD2.G (envelope plasmid), psPAX2 (packaging plasmid), and the pCDH-copGFP expression vector containing His-3xFLAG-FAT10 (FLAG-FAT10) (or the abovementioned variants FLAG-FAT10 A or FLAG-FAT10 E) in the ratio pMD2.G:psPAX2:expression plasmid of 1:1.84:2.1. The following day, cells were visually checked for syncytia and the medium was replaced with standard DMEM medium and incubated overnight. The lentiviral vectors were harvested 48–72 h after transfection. For harvesting the lentiviruses, the supernatants were collected, centrifuged at 4°C, 300$g$ for 5 min, and directly sterile-filtered through 0.45 $\mu$m filters to remove contaminating cells. The lentiviruses were either used directly to transduce the target cells or stored at −80°C. DNase digest was performed to remove remaining plasmid DNA in the lentiviral supernatant by mixing 1 $\mu$g/ml DNase and 1 mM MgCl$_2$ gently and incubating the mix at 37°C for 20 min.

### Transduction of cells using lentiviral vectors

Target cells were seeded at a density of 2 × 10$^6$ in a 100-mm dish and 5 ml of lentiviral suspension was added to the dish at a multiplicity of infection of 50 (MOI: 50). The cells were then incubated at 37°C for 48–72 h. After the incubation, the transduced cells were selected by GFP-positive sorting using BD FACS Aria Ilu cell sorter. The selected cells were further expanded and used for experiments.

### Induction of endogenous FAT10

Induction of endogenous FAT10 expression was performed using the above described medium with the addition of 300 U/ml human IFN$\gamma$ (300-02; Peprotech) and 600 U/ml human TNF (300-01A; Peprotech), as described earlier (Aichem et al, 2019). The induction was performed for 24 h to achieve the highest level of endogenous FAT10 expression.

### Induction and inhibition of FAT10 phosphorylation

Induction of FAT10 phosphorylation was performed using 600 U/ml human TNF for 24 h. When indicated, the cell lysate was exposed to 5–10 units of CIP phosphatase (0290; NEB) for 2 h at 37°C. When indicated, cells were pretreated for 3 h with the following inhibitors: 10 $\mu$M IKK-16 (pan-IKK inhibitor, S2882; Selleckchem), 10 $\mu$M SP600125 (pan-JNK inhibitor, S1460; Selleckchem), 10 $\mu$M SB203580 (p38 MAPK inhibitor, S1076; Selleckchem), 10 $\mu$M trametinib (MEK1/2 inhibitor, S2673; Selleckchem), 5 $\mu$M TPCA-1 (IKK$\beta$ inhibitor, SC-203083; Santa Cruz) or 10 $\mu$M CAY10576 (IKK$\varepsilon$ inhibitor, 10011249; Cayman). Induction of FAT10 phosphorylation was also performed by infecting A549 cells with IAV strain A/Regensburg/D6/09 (H1N1pdm09; RB1) at an MOI of 1.

### Confocal microscopy and live cell imaging

A549, A549 FLAG-FAT10 wt, A549 FLAG-FAT10 A, and A549 FLAG-FAT10 A cells were seeded on glass slices in 12-well plates to a confluency of 25% and cultured at 37°C and 5% CO$_2$. After 24 h, cells were washed with PBS and fixed with 4% formaldehyde in PBS for 10 min. Cells were washed twice in PB buffer (3% BSA; PBS) and permeabilized for 5 min with 0.2% Triton-X100/PBS. After two washing steps, the cells were incubated with primary antibodies for 2 h. Cells were washed three times with PBS buffer and incubated for 2 h with fluorescent-dye-coupled secondary antibodies, followed by three PBS buffer-washing steps. Images were acquired with a 63x Plan Apochromat oil objective using a Zeiss LSM700 confocal microscope.

## Supplementary Information

## Acknowledgements

We gratefully acknowledge Prof. Dr. Marcus Groettrup, who sadly passed away during the course of this study. This work was supported by the German Research Foundation (DFG) Collaborative Research Center CRC969, projects TP C01 and C09. H Urlaub is supported by the German Research Foundation (DFG, SFB1286). We acknowledge Dr. Silke Wiesner for the modeling of the FAT10 structure in Fig 1B, Hannah Honner for technical help, and Monika Raabe for assistance in mass spectrometry. We thank Prof. Daniel Pinschewer for contributions of VSV and VSV-GFP, and Prof. Oliver Planz for IAV strains.

### Author Contributions

K Saxena: formal analysis, validation, investigation, and methodology.
ND Roverato: conceptualization, formal analysis, investigation, methodology, and writing—original draft and project administration.
M Reithmann: investigation.
MM Mah: formal analysis, validation, investigation, and methodology.
R Schregle: investigation.
G Schmidtke: investigation.
I Silbern: investigation.
H Urlaub: formal analysis, supervision, funding acquisition, investigation, and methodology.
A Aichem: supervision, funding acquisition, investigation, project administration, and writing—review and editing.

### Conflict of Interest Statement

The authors declare that they have no conflict of interest.

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
