## [Reviewer comments · Life Science Alliance]

Life Science Alliance

FAT10 is phosphorylated by IKK β to inhibit the antiviral type-I interferon response

Kritika Saxena, Nicola Roverato, Melody Reithmann, Mei Mah, Richard Schregle, Gunter Schmidtke, Ivan Silbern, Henning Urlaub, and Annette Aichem

DOI: <https://doi.org/10.26508/lsa.202101282>

Corresponding author(s): Annette Aichem, Biotechnology Institute Thurgau

Review Timeline:

Submission Date:	2021-10-29
Editorial Decision:	2021-12-06
Revision Received:	2023-09-19
Editorial Decision:	2023-10-27
Revision Received:	2023-10-30
Accepted:	2023-10-30

Transaction Report:

December 6, 2021

Re: Life Science Alliance manuscript #LSA-2021-01282-T

Prof. Marcus Groettrup
University of Konstanz
Department of Biology, Division of Immunology
Universitätsstrasse 10
Universitaetsstrasse 10
Konstanz 78457
GERMANY

Dear Dr. Groettrup,

Thank you for submitting your manuscript entitled "FAT10 is phosphorylated by IKK β to limit the antiviral type-I interferon response". The manuscript has been evaluated by expert reviewers, whose reports are appended below. Unfortunately, after an assessment of the reviewer feedback, our editorial decision is against publication in Life Science Alliance.

Although your manuscript is intriguing, I feel that the points raised by the reviewers are more substantial than can be addressed in a typical revision period. If you wish to expedite publication of the current data, it may be best to pursue publication at another journal.

Given the interest in the topic, I would be open to resubmission to Life Science Alliance of a significantly revised and extended manuscript that fully addresses the reviewers' concerns and is subject to further peer-review. If you would like to resubmit this work to Life Science Alliance, you may submit an appeal directly through our manuscript submission system. Please note that priority and novelty would be reassessed at resubmission.

Regardless of how you choose to proceed, we hope that the comments below will prove constructive as your work progresses. We would be happy to discuss the reviewer comments further once you've had a chance to consider the points raised in this letter.

Thank you for thinking of Life Science Alliance as an appropriate place to publish your work.

Sincerely,

Reviewer #1 (Comments to the Authors (Required)):

In this study, the authors investigated the phosphorylation of FAT10. Their proteomics analysis revealed the phosphorylation sites of FAT10. They claimed that phosphorylation was induced by TNF-alpha, and IKK-beta is important for the phosphorylation. Influenza A virus infection potentiated FAT10 phosphorylation. They used phospho-defective and -mimetic FAT10 and found that those mutations did not affect the stability and subcellular localization of FAT10. They claimed that FAT10 phosphorylation was required for FAT10-mediated suppression of type I IFN expression, and OTUB1 was crucial for this phospho-FAT10-mediated inhibition of type I IFN response. Although this manuscript contains several interesting findings, but some of their data seem to be contradictory and not convincing. In addition, the phenotype of OTUB1 KO cells in this manuscript is different from those reported in a previous study (Cell Rep. 30: 1570-1584). Specific comments are described below.

Major comment 1:

In Fig 2, They showed that TNF-alpha and IFN-gamma treatment increased the amount of exogenously expressed FAT10. The authors should explain the reason why exogenous FAT10 levels were increased by the treatment, because they also claimed that phosphorylation did not affect the protein stability. Since FAT10 levels are different among the lanes (Fig 2B and 2C), it is unclear whether the treatment induced the phosphorylation or it increased the amounts of both non-phosphorylated and phosphorylated proteins. The authors should compare the phosphorylation of samples in which FAT10 levels were comparable.

Major comment 2

CIP treatment only slightly reduced the slow migrating bands (Fig 2B). The authors should provide more convincing data to

show CIP treatment eliminate the slow migrating bands. Additionally, the two lanes were split in Fig 2D, and thus lane 1 cannot be considered to be a control. The authors should show the data that does not contain the split.

Major comment 3:

In Fig 4F, the FAT10 level in lane 2 was comparable to that of lane 3 in "Load" blot, but the FAT10 level in lane 2 of low contrast bot was faint compared to that in lane 3. It is unclear why there is a large difference.

Major comment 4:

A previous study has shown that knockout of OTUB1 markedly reduces RIG-I-dependent IRF3 activation essential for type I IFN expression (Cell Rep. 30: 1570-1584), but Fig 7E showed that OTUB1 knockout exhibited marginal effect on type I IFN production in response to poly I:C. The authors obtained the OTUB1 KO cells from Sumana Sanyal who reported the above paper, Cell Rep. 30: 1570-1584. If OTUB1 is required for type I IFN production, it is opposite to the function of FAT10. If knockout of OTUB1 does not show significant effect on RIG-I-mediated type I IFN production, it is difficult to explain the underlying mechanism of how FAT10 regulates RIG-I activity via OTUB1. The authors should provide data to reconcile this apparent discrepancy.

Major comment 5:

Although the authors described "FAT10 does not sequester RIG-I into the insoluble fraction during IAV infection in contrast to a previous report", the experimental condition of this study is different from the previous study, and thus it seems to be inappropriate.

Major comment 6:

In this study, many of the experiments were performed using over-expression. Thus, it is difficult to assess the physiological significance of phosphorylation of FAT10. To assess the physiological significance of FAT10-mediated OTUB1 regulation, the authors should use double KO of FAT10 and OTUB1 in their experiments.

Minor comment 1:

In Fig 1C, molecular weight marker positions should be show in each IB, because it is unclear which bands in autoradiography are FAT10.

Minor comment 2

In Fig 7b, the label of y-axis, activated RIG-I, is confusing. It should be corrected, such as "RIG-I expression level".

Reviewer #2 (Comments to the Authors (Required)):

In this manuscript, the authors raises important finding that FAT10 can be phosphorylated and the consequence of its phosphorylation. Below are some comments and suggestions for the authors to consider

- 1) Lines 125 - 126, "In parallel, we performed a similar experiment with overexpressed FLAG-FAT10.": Ambiguous if this exogenous FAT10 expression was also performed in the presence of TNF α and IFN γ like in the previous experiment.
- 2) Lines 130 - 131: What is the methodology behind the screening of kinases phosphorylating FAT10? For example, what is the repertoire of kinases tested? 10 select kinases were shown in Figure EV1, were these the kinases with the highest activity detected in phosphorylating FAT10?
- 3) Certain lines might be misleading readers to believe FAT10 is only phosphorylated under inflammatory cytokine treatment, when it appears exogenous FLAG-FAT10 is also basally phosphorylated to a certain degree. Suggestion to say instead that phosphorylation of FAT10 is enhanced under TNF α treatment.
 - a. Evidence supporting basal phosphorylation of exogenously transfected FAT10:
 - i. Lines 126-128: "Remarkably, we found that FAT10 is phosphorylated at 5 different sites: Ser62, Ser64, Thr77, Ser95, Ser109 (Fig 1B), 128 with both endogenous and FLAG-tagged FAT10s displaying a similar phosphorylation profile
 - ii. Figure 2A, lane 2 - phosphorylated FAT10 was also present as a faint band when only FLAG-FAT10 is present in the absence of TNF/IFN. This is curtailed under CIP treatment (Lane 3).
 - b. Misleading lines:
 - i. Lines 137 - 138: "As the previous phospho-mass spectrometric approach indicated that FAT10 is phosphorylated under endogenous conditions..."
 - ii. Lines 148 - 149 : "Surprisingly, we found that FLAG-FAT10 phosphorylation was induced by TNF α /IFN γ ..."
 - iii. Line 154 - 155: "Thereby we confirmed that only TNF α /IFN γ treatment could induce p-FAT10 formation, which corresponds to approx. 5% of the total FAT10 protein amount"

Reviewer #3 (Comments to the Authors (Required)):

This paper demonstrates phosphorylation of FAT10 and its potential involvement in the regulation of antiviral signaling

pathways. It is of interesting observation, proposing the possible crosstalk between the virus induced interferon production pathways and the inflammatory cytokine (TNF)-mediated signaling pathways, via phosphorylated form of FAT10. Authors address whether FAT10 is phosphorylated, how this phosphorylation controlled, and what is the significances of this modification. It seems clear that FAT10 is indeed phosphorylated and IKK is the responsible kinase (in the case of TNF stimulation), however, I am not convinced mechanistically how the phosphorylated FAT10 controls IFN production. Although authors report many novel observations, those findings are somewhat distractive and I feel that the logical cohesiveness are rather weak.

Fig.1A. It needs more explanation. Experimentally, it is incomplete dataset. By itself, not informative. Control is missing and need verification of FAT10 (by immunoblot).

Fig.1C. What is the basis of choosing those tested kinases (in the extended data)? No explanation about them. It seems MK3 phosphorylates FAT10 in vitro. So, what? Data stops here and does not address further. If authors wish to show MK3-mediated phosphorylation of FAT10 is indeed meaningful, it is better to address its regulation and significance in more detail experimentally. Otherwise, it is better to show in vitro phosphorylation of FAT10 by IKK.

Fig2 How about phosphorylation of Flag-FAT10 at basal condition (compare lane 2/3 of Fig.2A)? Are they differ (or same) compared to phosphorylation induced ones by TNF? Or by overexpression of IKK kinases? Why Phos-taq SDS-PAGE does not recognize them? Is it specificity issue? Or sensitivity issue?

Figure 3

3C. Again, is it specific to TNF? Or is it still same for basal phosphorylation?

Fig.4

Line 228~. Hard to understand the logic here. Speculation and hypothesis mixed and not clearly stated. Experimentally, overexpress FAT10 and then treat TNF or infect with IAV. Which physiological situation this experiment is intending to mimic?

Fig 4E. Too preliminary and not consistent to previous data set. Here, recombinant form of FAT10 already possess phosphorylated forms (lane 2). Why? And lysate incubation alone reduces phosphorylation (lane 3). How? It is clear IAV infection affects phosphorylation of FAT10, but this dataset remains so many unanswered questions. Those has to be thoroughly addressed.

Fig5&6. To me, it looks both FAT10 A and FAT10 E mutants similarly lowers phosphorylation of FAT10(Fig. 5A). Yet, their effects on IFN β secretion differs (Fig 6 A-D). There are so many phospho bands here (Fig 5A). Specify them. If authors claim that phosphorylation status of FAT10 A and FAT10 E differ compare to WT, then quantify them. Need more reliable data. Fig.5C need quantification, too. Fig5D. I do not know what they are looking for. Need explanations.

Fig. 7. Similarly, I see different effect of FAT10 A and E in the data of 7C and 7D. It seems FAT10E behave similarly with WT FAT10, while FAT10A is differ. Yet, why compare FAT10E instead of FAT10A in 7E? The data on OTUB1 KO cells might be interesting. However, I do not understand what they meant, in terms of role of FAT10 phosphorylation. In general, role of OTUB1 in the phosphorylated FAT10 mediated IFN regulation is not conclusive.

Fig7G. So, what is the molecular role of phosphorylated FAT10? In this model, it seems authors proposing that phosphorylated form of FAT10 preferentially interact with OTUB1. However, I do not see dataset supporting this claim. In Fig7C. Both wt and FAT10 E similarly binds to OTUB1. There is no TNF or kinase to phosphorylate FAT10 in this data set. How this can be explained in Fig7G model??

Point-by-point reply for manuscript #LSA-2021-01282-T

We would like to thank all our reviewers for their constructive criticism. We are convinced that our manuscript has greatly benefited from their comments and suggestions. For a better orientation, we have highlighted all main text changes in yellow.

The mass spectrometry proteomics data have been deposited to the ProteomeXchange Consortium via the PRIDE partner repository with the dataset identifier PXD028918.
(Username: reviewer_pxd028918@ebi.ac.uk, Password: IWQfsUtV)

Reviewer #1 (Comments to the Authors (Required)):

In this study, the authors investigated the phosphorylation of FAT10. Their proteomics analysis revealed the phosphorylation sites of FAT10. They claimed that phosphorylation was induced by TNF-alpha, and IKK-beta is important for the phosphorylation. Influenza A virus infection potentiated FAT10 phosphorylation. They used phospho-defective and -mimetic FAT10 and found that those mutations did not affect the stability and subcellular localization of FAT10. They claimed that FAT10 phosphorylation was required for FAT10-mediated suppression of type I IFN expression, and OTUB1 was crucial for this phospho-FAT10-mediated inhibition of type I IFN response. Although this manuscript contains several interesting findings, but some of their data seem to be contradictory and not convincing. In addition, the phenotype of OTUB1 KO cells in this manuscript is different from those reported in a previous study (Cell Rep. 30: 1570-1584). Specific comments are described below.

Major comment 1:

In Fig 2, They showed that TNF-alpha and IFN-gamma treatment increased the amount of exogenously expressed FAT10. The authors should explain the reason why exogenous FAT10 levels were increased by the treatment, because they also claimed that phosphorylation did not affect the protein stability. Since FAT10 levels are different among the lanes (Fig 2B and 2C), it is unclear whether the treatment induced the phosphorylation or it increased the amounts of both non-phosphorylated and phosphorylated proteins. The authors should compare the phosphorylation of samples in which FAT10 levels were comparable.

We are aware of the increased expression of the FLAG-FAT10 protein in presence of TNF. For this reason, we already commented in the text that the activation of the TNF signaling pathway is known to lead by itself to the activation of the CMV promoter, which is upstream of the FLAG-FAT10 gene (lines 148-151). In Fig 2C the amount of FLAG-FAT10 in lanes 4 and 5 is indeed slightly increased as compared to lanes 2 and 3, however, no phosphorylation at all is detectable in lanes 2 and 3. We think that if the signals of phospho-FAT10 detected in lanes 4 and 5 would result just from a stronger expression of FLAG-FAT10, at least a slight FAT10 phosphorylation should be seen in lanes 2 and 3, as well.

Moreover, we have now repeated and exchanged the experiment shown in the former Fig 2B. As it can be seen in the new Fig 2B, FLAG-FAT10 amounts are equal and phosphorylation of FAT10 is mediated only in the presence TNF/IFN γ (Fig 2B, lane 4).

Major comment 2

CIP treatment only slightly reduced the slow migrating bands (Fig 2B). The authors should provide

more convincing data to show CIP treatment eliminate the slow migrating bands. Additionally, the two lanes were split in Fig 2D, and thus lane 1 cannot be considered to be a control. The authors should show the data that does not contain the split.

We completely agree with our reviewer. We assume that the CIP phosphatase could only partially de-phosphorylate FAT10, due to an intrinsic low affinity for proteins. For our revision, we have repeated the experiment in Fig 2B using lambda phosphatase, which is better known to remove phosphate groups from proteins. New Fig 2B shows now a complete reduction of the FAT10 phosphorylation upon treatment with lambda phosphatase (Fig 2B, lane 5). Moreover, we have repeated FAT10 phosphorylation under endogenous conditions upon up-regulation of endogenous FAT10 expression with IFN γ and TNF (new Fig 2B, lanes 6 and 7). Here, phosphorylated FAT10 can now be seen directly side-by-side with untreated cells and FLAG-FAT10 expressing cells.

Major comment 3:

In Fig 4F, the FAT10 level in lane 2 was comparable to that of lane 3 in "Load" blot, but the FAT10 level in lane 2 of low contrast bot was faint compared to that in lane 3. It is unclear why there is a large difference.

We completely agree with the observation of our reviewer. We have repeated and replaced this figure. We show now an experiment with equal FLAG-FAT10 amounts in each lane. As can be seen in new Fig 4F, phosphorylation is increased upon IAV infection (Fig 4F, lane 2 versus lane 3). When cells were treated at the same time with an IKK β inhibitor, FAT10 phosphorylation was diminished, but not when cells were treated with an IKK ϵ inhibitor (Fig 4F, lanes 4 and 5).

Major comment 4:

A previous study has shown that knockout of OTUB1 markedly reduces RIG-I-dependent IRF3 activation essential for type I IFN expression (Cell Rep. 30: 1570-1584), but Fig 7E showed that OTUB1 knockout exhibited marginal effect on type I IFN production in response to poly I:C. The authors obtained the OTUB1 KO cells from Sumana Sanyal who reported the above paper, Cell Rep. 30: 1570-1584. If OTUB1 is required for type I IFN production, it is opposite to the function of FAT10. If knockout of OTUB1 does not show significant effect on RIG-I-mediated type I IFN production, it is difficult to explain the underlying mechanism of how FAT10 regulates RIG-I activity via OTUB1. The authors should provide data to reconcile this apparent discrepancy.

We would like to thank our reviewer for this comment. Jahan et al. (2020) used A549 cells for their experiments, as we do. However, Sanyal and colleagues infected A549 cells with IAV and showed IRF3 activation by immunoblot analysis, rather than type-I IFN production by ELISA. Here, we assessed the levels of IFN β after Poly (I:C) transfection, which represents a completely different experimental setup. However, we have made the observation that the measured IFN β levels differ a lot if the cells are treated either with Poly (I:C), or if they are infected with IAV, as can be seen in the figure below. While we always observed a clear decrease in IFN β secretion in A549 OTUB1-KO cells infected with IAV (left panel, A), the secreted IFN β levels remained high when the cells were transfected with Poly (I:C) (right panel, B). As a further proof, we have performed the experiment shown in Fig. 7E now also with IAV infected cells (new Figure 7F). In both cases, upon Poly (I:C) transfection and IAV infection, the expression of FAT10 has a clear negative impact on IFN β secretion, and in both cases, this effect is abolished in OTUB1-KO cells. Unfortunately, we cannot explain what causes the difference in Poly (I:C) or IAV treated cells. However, since the negative impact of FAT10 is the same and since in both cases the OTUB1-KO abolished this effect, we believe that our data strongly support our hypothesis that phosphorylated FAT10 mediates the negative impact on IFN β secretion via its interaction with OTUB1. We hope that the new data will now also convince our reviewer.

Major comment 5:

Although the authors described "FAT10 does not sequester RIG-I into the insoluble fraction during IAV infection in contrast to a previous report", the experimental condition of this study is different from the previous study, and thus it seems to be inappropriate.

In this study, we have investigated the sequestration of endogenous RIG-I into the insoluble fraction upon overexpression of FLAG-FAT10. However, we agree that the experimental condition of this study is different from the previous study and we think that the data obtained under endogenous conditions are more reliable than the data obtained under overexpressing conditions by Nguyen et al. (2016) *Sci. Rep.* 6:23377. Therefore, we have now repeated the experiment in new Fig S4B under fully endogenous conditions (endogenous RIG-I and endogenous FAT10 in wild type and FAT10 KO cells), to further strengthen our finding. We show these data along with the FLAG-FAT10 overexpression data in new Fig S4. In both cases, we did not observe any significant accumulation of RIG-I in the insoluble fraction.

Major comment 6:

In this study, many of the experiments were performed using over-expression. Thus, it is difficult to assess the physiological significance of phosphorylation of FAT10. To assess the physiological significance of FAT10-mediated OTUB1 regulation, the authors should use double KO of FAT10 and OTUB1 in their experiments.

We completely agree with this interesting suggestion. We have now performed the experiments shown in Figure 7E and new Figure 7F under completely endogenous conditions, upon stimulation of endogenous FAT10 expression with TNF/IFN γ combined with Poly (I:C) or IAV treatment. As can be seen in the new Figures 7G and 7H, induction of endogenous FAT10 expression likewise down-regulated IFN β secretion in wild type A549 cells, and this was likewise abolished in A549 FAT10 KO, OTUB1 KO and FAT10/OTUB1 double knockout cells. We are thankful for this suggestion, since these data strongly support our data shown in Figure 7E and 7F, where we have worked with overexpressed FAT10 variants.

Minor comment 1:

In Fig 1C, molecular weight marker positions should be shown in each IB, because it is unclear which bands in autoradiography are FAT10.

We have now inserted the molecular weight markers in the blots, as suggested.

Minor comment 2

In Fig 7b, the label of y-axis, activated RIG-I, is confusing. It should be corrected, such as "RIG-I expression level".

We agree and have re-labeled the y-axis as "Total RIG-I (% of WT) (Normalized to GAPDH)"

Reviewer #2 (Comments to the Authors (Required)):

In this manuscript, the authors raise an important finding that FAT10 can be phosphorylated and the consequence of its phosphorylation. Below are some comments and suggestions for the authors to consider

1) Lines 125 - 126, "In parallel, we performed a similar experiment with overexpressed FLAG-FAT10.": Ambiguous if this exogenous FAT10 expression was also performed in the presence of TNF α and IFN γ like in the previous experiment.

We would like to thank our reviewer for this comment. The FLAG-FAT10 phospho-proteomic experiments were performed in the absence of TNF, which means that there indeed exists a basal level of phosphorylated FAT10, which has been detected with the very sensitive mass spectrometric analysis. Nevertheless, we show that FAT10 phosphorylation is further increased by TNF treatment (Fig 2B and 2C) and upon IAV infection (Fig 4). We mention this now in the text (line 110 ff): "In parallel, we performed a similar experiment with overexpressed 6His-3xFLAG-FAT10 (named hereafter as FLAG-FAT10), however, in this case in the absence of TNF."

2) Lines 130 - 131: What is the methodology behind the screening of kinases phosphorylating FAT10? For example, what is the repertoire of kinases tested? 10 select kinases were shown in Figure EV1, were these the kinases with the highest activity detected in phosphorylating FAT10?

We agree that the entire data set of the screen of 256 Ser/Thr kinases should be shown in the Supplementary data. In this large original screen the recombinant kinases were incubated with recombinant FAT10 and incorporation of radioactive phosphate was monitored. The 10 kinases shown in Fig S1 were selected based, both, on prominent phosphorylation, and biological significance related to FAT10 function. We have now added the data of the original screen of the recombinant kinases as Supplementary Table S2.

3) Certain lines might be misleading readers to believe FAT10 is only phosphorylated under inflammatory cytokine treatment, when it appears exogenous FLAG-FAT10 is also basally phosphorylated to a certain degree. Suggestion to say instead that phosphorylation of FAT10 is enhanced under TNF α treatment.

We agree with our reviewer. We have changed the text that and write now that a basal FAT10 phosphorylation occurs but is enhanced upon TNF treatment:

Line 142 ff: "Since our phospho-proteomic analysis had revealed that FLAG-FAT10 was phosphorylated already in the absence of TNF/IFN γ , we suggest that a basal phosphorylation of FAT10 must exist which is further enhanced by TNF/IFN γ treatment."

a. Evidence supporting basal phosphorylation of exogenously transfected FAT10:

i. Lines 126-128: "Remarkably, we found that FAT10 is phosphorylated at 5 different sites: Ser62, Ser64, Thr77, Ser95, Ser109 (Fig 1B), 128 with both endogenous and FLAG-tagged FAT10s displaying a similar phosphorylation profile

ii. Figure 2A, lane 2 - phosphorylated FAT10 was also present as a faint band when only FLAG-FAT10 is present in the absence of TNF/IFN. This is curtailed under CIP treatment (Lane 3).

We completely agree and have changed the text as shown above (line 140ff).

b. Misleading lines:

i. Lines 137 - 138: "As the previous phospho-mass spectrometric approach indicated that FAT10 is phosphorylated under endogenous conditions..."

We have changed the text in line 122ff as follows:

"As our phospho-mass spectrometric approach indicated that FAT10 is phosphorylated both, under endogenous conditions in TNF/IFN γ stimulated HEK293 cells, as well as in the absence of cytokines, we aimed to corroborate this notion by using a combined immunoprecipitation (IP)/immunoblot (IB) approach."

ii. Lines 148 - 149 : "Surprisingly, we found that FLAG-FAT10 phosphorylation was induced by TNF α /IFN γ ..."

We have changed the text in line 132ff as follows:

"Interestingly, we found that phosphorylation of FLAG-FAT10 was enhanced by TNF/IFN γ , and that this modification was strongly reversed by CIP treatment (Fig 2A, lanes 4 and 5).

iii. Line 154 - 155: "Thereby we confirmed that only TNF α /IFN γ treatment could induce p-FAT10 formation, which corresponds to approx. 5% of the total FAT10 protein amount"

We have changed the text in line 142ff as follows:

"Since our phospho-proteomic analysis had revealed that FLAG-FAT10 was phosphorylated already in the absence of TNF/IFN γ , we suggest that a basal phosphorylation of FAT10 must exist which is further enhanced by TNF/IFN γ treatment. Of note, the portion of FAT10 which becomes phosphorylated was estimated to be approximately 5% of the total FAT10 protein amount."

Reviewer #3 (Comments to the Authors (Required)):

This paper demonstrates phosphorylation of FAT10 and its potential involvement in the regulation of antiviral signaling pathways. It is of interesting observation, proposing the possible crosstalk between the virus induced interferon production pathways and the inflammatory cytokine (TNF)-mediated signaling pathways, via phosphorylated form of FAT10. Authors address whether FAT10 is phosphorylated, how this phosphorylation controlled, and what is the significances of this modification. It seems clear that FAT10 is indeed phosphorylated and IKK is the responsible kinase (in the case of TNF stimulation), however, I am not convinced mechanistically how the phosphorylated FAT10 controls IFN production. Although authors report many novel observations, those findings are somewhat distractive and I feel that the logical cohesiveness are rather weak.

Fig.1A. It needs more explanation. Experimentally, it is incomplete dataset. By itself, not informative. Control is missing and need verification of FAT10 (by immunoblot).

We thank our reviewer for this comment. As a control, we have used untreated HEK293 cells, since FAT10 is expressed only in presence of TNF/IFN γ . Thus, no additional controls can be included in this experiment. We have exchanged Fig 1A and have now included also the respective immunoblot analysis of endogenous FAT10 induction and conjugation. As can be seen in new Fig 1A and B, the FAT10 band appears only in presence of TNF α /IFN γ (Fig 1A and B, lane 2).

Fig.1C. What is the basis of choosing those tested kinases (in the extended data)? No explanation about them. It seems MK3 phosphorylates FAT10 *in vitro*. So, what? Data stops here and does not address further. If authors wish to show MK3-mediated phosphorylation of FAT10 is indeed meaningful, it is better to address its regulation and significance in more detail experimentally. Otherwise, it is better to show *in vitro* phosphorylation of FAT10 by IKK.

MK3 was shown to phosphorylate recombinant FAT10 under *in vitro* conditions (Fig 1C). However, we also show that MK3 does not act as a FAT10 kinase *in cellulo* (Fig 2A). Accordingly, we did not claim any biological function for MK3 as a FAT10 kinase. Nevertheless, we think that the MK3-mediated robust phosphorylation of recombinant FAT10 could be used as an informative positive control in order to validate our *in vitro* experiments on FAT10 phosphorylation. We mention now in the text that MK3 functions only under *in vitro* conditions, however not *under in cellulo* conditions: Line 134 ff: “However, neither MK3 overexpression, nor starvation/TPA treatment could enhance the phosphorylation of FAT10 under in cellulo conditions (Fig 2A, lanes 6-9).” Moreover, later in the text (line 182 ff) we additionally mention that “We performed an *in vitro* kinase assay using recombinant FAT10 (rFAT10), rIKK β , rIKK ϵ and rMK3, (the latter was utilized as a positive control),..”.

Fig2 How about phosphorylation of Flag-FAT10 at basal condition (compare lane 2/3 of Fig.2A)? Are they differ (or same) compared to phosphorylation induced ones by TNF? Or by overexpression of IKK kinases? Why Phos-taq SDS-PAGE does not recognize them? Is it specificity issue? Or sensitivity issue?

We apologize that we did not explain clearly enough that we also observed and measured a basal phosphorylation of FAT10. We have now changed the text accordingly and hope to make this point more clear for the reader: lines 110ff , 122ff and 143ff.

Interestingly, by immunoblot analysis, we always detected basal FAT10 phosphorylation in A549 cells (for example in new Fig 5A), while we were never able to detect basal FAT10 phosphorylation in HEK293 cells, showing that the detection of basal FAT10 phosphorylation depends also on the cell type.

Figure 3

3C. Again, is it specific to TNF? Or is it still same for basal phosphorylation?

In Figure 3C lane 3, it is possible to see the TNF-induced phosphorylation of FAT10 at a high contrast (middle panel) which is not observed when FLAG-FAT10 is expressed without TNF stimulation (lane 2). In this experiment we show that the overexpression of the indicated kinases is strongly enhancing FAT10 phosphorylation.

Fig.4

Line 228~. Hard to understand the logic here. Speculation and hypothesis mixed and not clearly

stated. Experimentally, overexpress FAT10 and then treat TNF or infect with IAV. Which physiological situation this experiment is intending to mimic?

FAT10 is expressed when TNF and IFN γ are present in the micro-environment. IAV infection in the lung triggers the inflammation-induced production of these two cytokines, inducing FAT10 expression (Mah et. al, 2019). Once FAT10 expression has been induced, TNF and IAV are able to enhance FAT10 phosphorylation, with has direct consequences for the production of type-I interferons.

Fig 4E. Too preliminary and not consistent to previous data set. Here, recombinant form of FAT10 already possess phosphorylated forms (lane 2). Why? And lysate incubation alone reduces phosphorylation (lane 3). How? It is clear IAV infection affects phosphorylation of FAT10, but this dataset remains so many unanswered questions. Those has to be thoroughly addressed.

We completely agree with our reviewer. We have repeated the experiment and show now in new Fig 4E very clearly that recombinant FAT10 is phosphorylated when incubated with cell lysate of IAV-infected A459 cells, but not when the cells were not infected with IAV.

Fig5&6. To me, it looks both FAT10 A and FAT10 E mutants similarly lowers phosphorylation of FAT10(Fig. 5A). Yet, their effects on IFN β secretion differs (Fig 6 A-D). There are so many phospho bands here (Fig 5A). Specify them. If authors claim that phosphorylation status of FAT10 A and FAT10 E differ compare to WT, then quantify them. Need more reliable data. Fig.5C need quantification, too. Fig5D. I do not know what they are looking for. Need explanations.

We agree on the need for quantification of the phosphorylation status of FAT10 in this experiment. We have repeated the experiment now three times in A549 cells (new Fig 5A) and we show in the quantification in new Fig 5B that phosphorylation of the FAT10 A mutant is significantly diminished, as compared to wild type FLAG-FAT10. Since the mutation of the phosphorylation sites to glutamic acid already causes a retardation of FAT10 E proteins in Phos-tag gels, we cannot make a statement about the phosphorylation status of FAT10 E. We point to this now also in the text (line 241ff).

Moreover, we have also quantified the CHX chase data from three independent experiments as shown in Fig 5D and show the quantification in new Fig. 5E. Here it can be seen, that FLAG-FAT10, -A, and -E mutants are degraded with the same velocity.

The objective of Fig 5F was to show that mutations of the phosphorylation sites have no impact on the subcellular localization of FAT10. If this would be the case, we would not be able to directly compare the functional outcomes of these mutations.

Fig. 7. Similarly, I see different effect of FAT10 A and E in the data of 7C and 7D. It seems FAT10E behave similarly with WT FAT10, while FAT10A is differ. Yet, why compare FAT10E instead of FAT10A in 7E? The data on OTUB1 KO cells might be interesting. However, I do not understand what they meant, in terms of role of FAT10 phosphorylation. In general, role of OTUB1 in the phosphorylated FAT10 mediated IFN regulation is not conclusive.

In Fig 7E, we show that a knockout of OTUB1 is abolishing the negative effect that FAT10 has in inhibiting the type-I IFN response. Thus, we consider these data as a strong support to our conclusions. We compared FAT10 WT and FAT10 E since FAT10 A was already shown to have only a minor impact on type-I IFN secretion in Fig 6A and 6B, which address a similar question. By using FAT10 E and by seeing the same effect as with FAT10 WT we show that the inhibitory effect of FAT10 on IFN- β is indeed mediated by FAT10 phosphorylation. Moreover, we have now included additional experiments in new Fig 7F. Here we confirm that the inhibitory effect FAT10

has on IFN β secretion is also true upon IAV infection. In new figures 7G and H, this is further confirmed under completely endogenous conditions in A549 wild type cells, while it is not seen in FAT10 KO, OTUB1 KO cells or in FAT10-OTUB1 double knockout cells. We believe that all these data together provide strong evidence that FAT10 exerts its negative impact on IFN β secretion by its ability to bind stronger to OTUB1 when FAT10 is phosphorylated.

Fig7G. So, what is the molecular role of phosphorylated FAT10? In this model, it seems authors proposing that phosphorylated form of FAT10 preferentially interact with OTUB1. However, I do not see dataset supporting this claim. In Fig7C. Both wt and FAT10 E similarly binds to OTUB1. There is no TNF or kinase to phosphorylate FAT10 in this data set. How this can be explained in Fig7G model??

In Fig 7C, we show that FAT10 A has a reduced capacity to bind to OTUB1 as compared to FAT10 WT or FAT10 E. However, we do not claim that FAT10 A does not interact at all with OTUB1. Since we also detected a basal phosphorylation of FAT10, this might be enough to mediate binding of wild type FAT10 to OTUB1, and also to the same extent as observed with FAT10 E. Moreover, there is a strong correlation between the reduced binding of FAT10 A to OTUB1 and the ubiquitylation status of TRAF3 in Fig 7D. Together, these data are in line with the biological outcome we show in Fig 6 and 7E-H.

October 27, 2023

RE: Life Science Alliance Manuscript #LSA-2021-01282-TR-A

Dr. Annette Aichem
Biotechnology Institute Thurgau
Unterseestrasse 47
Kreuzlingen 8280
Switzerland

Dear Dr. Aichem,

Thank you for submitting your revised manuscript entitled "FAT10 is phosphorylated by IKK β to inhibit the antiviral type-I interferon response". We would be happy to publish your paper in Life Science Alliance pending final revisions necessary to meet our formatting guidelines.

- please address Reviewer 1's remaining comment
- please add a callout for Figure 5C to your main manuscript text
- you may want to consider uploading Figure 8 as a Graphical Abstract instead of as a figure, but this is up to you

A. FINAL FILES:

B. MANUSCRIPT ORGANIZATION AND FORMATTING:

****It is Life Science Alliance policy that if requested, original data images must be made available to the editors. Failure to provide**

original images upon request will result in unavoidable delays in publication. Please ensure that you have access to all original data images prior to final submission.**

The license to publish form must be signed before your manuscript can be sent to production. A link to the electronic license to publish form will be sent to the corresponding author only. Please take a moment to check your funder requirements.

Sincerely,

Reviewer #1 (Comments to the Authors (Required)):

The authors have addressed most of my previous concerns. They provided additional data showing that OTUB1 knockout phenotype is different in type I IFN response between poly I:C-stimulated and flu-infected cells. Since both short polyI:C and flu viral RNAs are recognized by RIG-I, the reason why there is a difference should be discussed.

Reviewer #2 (Comments to the Authors (Required)):

The authors have now addressed the concerns raised reasonably and it's now acceptable for publication

Reviewer #3 (Comments to the Authors (Required)):

This revised manuscript provides some answers to my previous concerns, but it is not satisfactory.

Point-by-point reply for manuscript #LSA-2021-01282-TR-A

We would like to thank our reviewers for their constructive criticism and we are convinced that our manuscript has clearly improved by the revision process.

Reviewer #1 (Comments to the Authors (Required)):

The authors have addressed most of my previous concerns. They provided additional data showing that OTUB1 knockout phenotype is different in type I IFN response between poly I:C-stimulated and flu-infected cells. Since both short polyI:C and flu viral RNAs are recognized by RIG-I, the reason why there is a difference should be discussed.

We are happy that our new data could satisfy our reviewer. We have now included a short text passage into the discussion (lines 396-404) where we point to the finding that we always observed a difference in the IFN β levels in Poly (I:C) or IAV infected A549 cells:

“It is interesting to note that the measured IFN β levels differed always markedly between A549 OTUB1 knockout cells which were treated either with Poly (I:C), or which were infected with IAV (Fig 7E and F). While we always observed a clear decrease in IFN β secretion upon infection with IAV in A549 OTUB1-KO cells as compared to wild type cells (black bars in Fig 7F), the secreted IFN β levels remained high in OTUB1 knockout cells which were transfected with Poly (I:C) (black bars in Fig 7E). This might be explained by the robustness of RIG-I activation caused by differences due to a transfection of Poly (I:C) as compared to RIG-I activation upon a viral infection. This might eventually result in the activation of other signaling pathways, contributing to IFN γ secretion. Of course, the exact mechanism causing this difference should be investigated in future experiments.”

Reviewer #2 (Comments to the Authors (Required)):

The authors have now addressed the concerns raised reasonably and it's now acceptable for publication

We are happy that our revised manuscript could satisfy our reviewer.

Reviewer #3 (Comments to the Authors (Required)):

This revised manuscript provides some answers to my previous concerns, but it is not satisfactory.

We are sorry that our reviewer is still not satisfied, although we have addressed all his concerns.

October 30, 2023

RE: Life Science Alliance Manuscript #LSA-2021-01282-TRR

Dr. Annette Aichem
Biotechnology Institute Thurgau
Unterseestrasse 47
Kreuzlingen 8280
Switzerland

Dear Dr. Aichem,

Thank you for submitting your Research Article entitled "FAT10 is phosphorylated by IKK β to inhibit the antiviral type-I interferon response". It is a pleasure to let you know that your manuscript is now accepted for publication in Life Science Alliance. Congratulations on this interesting work.

DISTRIBUTION OF MATERIALS:

Again, congratulations on a very nice paper. I hope you found the review process to be constructive and are pleased with how the manuscript was handled editorially. We look forward to future exciting submissions from your lab.

Sincerely,
